# From Language to Locomotion: Retargeting-free Humanoid Control via Motion Latent Guidance

**Zhe Li**[1][†][*], **Yangyang Wei**[2][*], **Boan Zhu**[3][*], **Yibo Peng**[1], **Tao Huang**[4], **Pengwei Wang**[1]
**Zhongyuan Wang**[1], **Cheng Chi**[1][✉], **Chang Xu**[6][✉], **Shanghang Zhang**[1,5][✉]
[1] BAAI, [2] Harbin Institute of Technology, [3] Hong Kong University of Science and Technology
[4] Shanghai Jiao Tong University, [5] Peking University, [6] University of Sydney

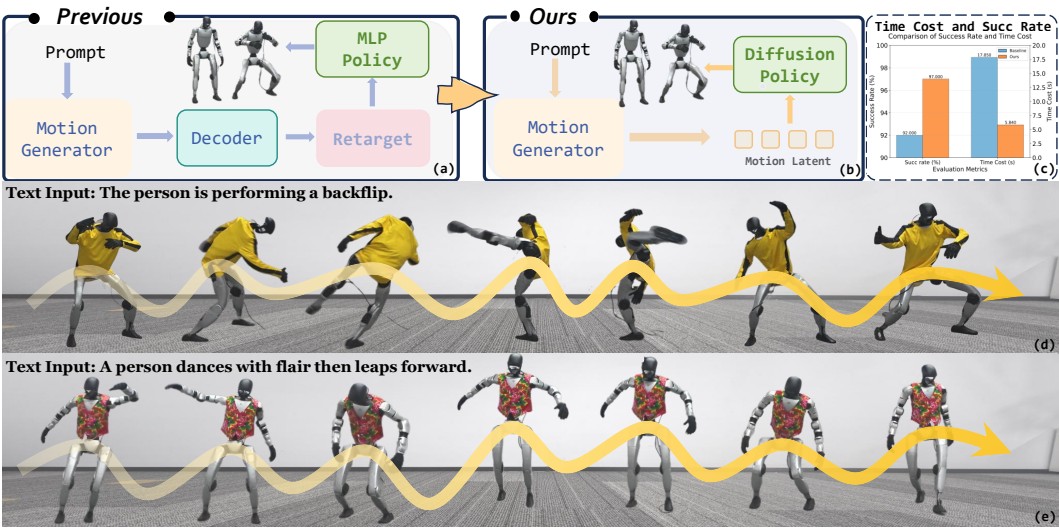

Figure 1: 🤖 **RoboGhost** is a retargeting-free latent driven policy for language-guided humanoid locomotion. By removing the dependency on motion retargeting, it thus allows robots to be controlled directly via open-ended language commands. The figure showcases (a) the previous pipeline with motion retargeting, (b) our proposed retargeting-free latent-driven pipeline, (c) quantitative comparisons of success rate and time cost between baseline and RoboGhost, (d) performing the backflip, and (e) dancing and leaping forward.

## Abstract

Natural language offers a natural interface for humanoid robots, but existing language-guided humanoid locomotion pipelines remain cumbersome and unreliable. They typically decode human motion, retarget it to robot morphology, and then track it with a physics-based controller. However, this multi-stage process is prone to cumulative errors, introduces high latency, and yields weak coupling between semantics and control. These limitations call for a more direct pathway from language to action, one that eliminates fragile intermediate stages. Therefore, we present **RoboGhost**, a retargeting-free framework that directly conditions humanoid policies on language-grounded motion latents. By bypassing explicit motion decoding and retargeting, RoboGhost enables a diffusion-based policy to denoise executable actions directly from noise, preserving semantic intent and supporting fast, reactive control. A hybrid causal transformer–diffusion motion generator further ensures long-horizon consistency while maintaining stability and diversity, yielding rich latent representations for precise humanoid behavior. Extensive experiments demonstrate that RoboGhost substantially reduces deployment latency, improves success rates and tracking accuracy, and produces smooth,

---

[*]Equal Contribution    [†] Project Leader    [✉] Corresponding Author

semantically aligned locomotion on real humanoids. Beyond text, the framework naturally extends to other modalities such as images, audio, and music, providing a general foundation for vision–language–action humanoid systems. *Project Page*

# 1 INTRODUCTION

Natural language provides an intuitive interface for humanoid robots, enabling users to translate free-form instructions into physically feasible humanoid motion. Recent text-to-motion (T2M) models can generate diverse and semantically meaningful human motions (Li et al., 2024d; Chen et al., 2023; Guo et al., 2024; Li et al., 2024b; Tevet et al., 2022). However, deploying these models on real robots typically requires a hierarchical pipeline: decoding human motion from language, retargeting it to robot morphology, and tracking the trajectory with a physics-based controller (Peng et al., 2018; Ji et al., 2024; Chen et al., 2025; He et al., 2025a; Zhou et al., 2025b).

This pipeline, while functional in constrained settings, suffers from systemic drawbacks. (1) *Errors accumulate* across decoding, retargeting, and tracking, degrading both semantic fidelity and physical feasibility. (2) *High latency* from multiple sequential stages, making real-time interaction difficult. (3) *Loose coupling* between language and control, since each stage is optimized in isolation rather than end-to-end. Recent refinements (Yue et al., 2025; Serifi et al., 2024; Shao et al., 2025) attempt to mitigate these issues, but improvements are applied locally to decoders or controllers, leaving the overall pipeline fragile and inefficient.

***Our key insight is simple: treat motion latents as first-class conditioning signals, directly use them as conditions to generate humanoid actions without decoding and retargeting altogether.*** We therefore propose ***RoboGhost***, a retargeting-free framework named to *highlight the latent representations that are invisible like a ghost yet strongly drive humanoid behavior*. As shown in Fig 1, rather than producing explicit human motion, RoboGhost leverages language-conditioned motion latents as semantic anchors to guide a diffusion-based humanoid policy. The policy denoises executable actions directly from noise, eliminating error-prone intermediate stages while preserving fine-grained intent and enabling fast, reactive control. To our knowledge, this is ***the first diffusion-based humanoid policy driven by motion latents***, achieving smooth and natural locomotion with DDIM-accelerated sampling (Song et al., 2020) for real-time deployment.

To enhance temporal coherence and motion diversity, RoboGhost employs a hybrid causal transformer–diffusion architecture. The transformer backbone captures long-horizon dependencies and ensures global consistency (Zhou et al., 2024; 2025a; Han et al., 2025). The diffusion component contributes stability and stochasticity for fine-grained motion synthesis. Together, this design mitigates drift and information loss typical of autoregressive models, while producing expressive motion latents that provide downstream policies with rich semantic conditioning for precise control.

Extensive experiments validate the effectiveness and practicality of RoboGhost. We dramatically accelerate the full pipeline from motion generation to humanoid deployment, cutting the time from $17.85\ s$ to $5.84\ s$. Beyond sheer speed, our approach yields higher-quality control by circumventing retargeting losses and improving generalization, which is reflected in a 5% higher success rate and reduced tracking error compared to baseline methods. Our framework achieves robust, semantically aligned locomotion on real humanoids, substantially reducing latency compared to retargeting-based pipelines. We can further extend this framework to support other input modalities such as images, audio, and music, thereby offering a reference architecture for humanoid vision-language-action systems. In short, ***RoboGhost moves text-driven humanoid control from fragile pose imitation to robust, real-time interaction.***

In summary, our key contributions can be summarized as follows:

- We propose RoboGhost, a retargeting-free framework that directly leverages language-generated motion latents for end-to-end humanoid policy learning, eliminating error-prone decoding and retargeting stages.

- We introduce the first diffusion-based humanoid policy conditioned on motion latents, which denoises executable actions directly from noise and achieves smooth, diverse, and physically plausible locomotion with DDIM-accelerated sampling.

- We design a hybrid transformer–diffusion architecture that unifies long-horizon temporal coherence with stochastic stability, yielding expressive motion latents and strong language–motion alignment.

- We validate RoboGhost through extensive experiments, demonstrating its effectiveness and generality in enabling robust and real-time language-guided humanoid locomotion.

## 2  RELATED WORK

### 2.1  HUMAN MOTION SYNTHESIS

Language-guided humanoid locomotion leverages advances in text-to-motion generation, which primarily uses transformer-based discrete modeling or diffusion-based continuous modeling. Discrete methods model motion as tokens, evolving from early vector quantization (Guo et al., 2022c) to GPT-style autoregression (Zhang et al., 2023a), scaling with LLMs (Jiang et al., 2023; Zhang et al., 2024) or improved attention (Zhong et al., 2023), and recently, bidirectional masking (Guo et al., 2023) and enhanced text alignment (Li et al., 2024d). In parallel, diffusion models excel in synthesis (Kim et al., 2023; Hu et al., 2023; Tevet et al., 2022; Li et al., 2024c; Bai et al., 2025; Li et al., 2023b), with progress in efficiency (Chen et al., 2023), retrieval-augmentation (Zhang et al., 2023b), controllability (Li et al., 2024b), and architectures (Meng et al., 2024; Li et al., 2025c). Our work builds on the continuous autoregressive framework (Li et al., 2024a), combining the benefits of both paradigms.

### 2.2  HUMANOID WHOLE-BODY CONTROL

Whole-body control (WBC)  (Li et al., 2025a;b) is crucial for humanoids, yet learning a general-purpose policy remains challenging. Existing approaches exhibit inherent trade-offs: methods like OmniH2O (He et al., 2024) and HumanPlus (Fu et al., 2024) prioritize specific robustness at the cost of generality or long-term accuracy, while others like Hover (He et al., 2025b), ExBody2 (Ji et al., 2024), and GMT (Chen et al., 2025) employ strategies (e.g., masking, curriculum learning, MoE) to enhance adaptability, though generalization is not guaranteed. Recent language-guided works also face limitations: LangWBC (Shao et al., 2025) scales poorly with no generalization guarantee to unseen instructions, RLPF (Yue et al., 2025) risks catastrophic forgetting and limited diversity, UH-1 (Mao et al., 2025) is a transformer-based large model relying on motion retargeting and discrete action tokenization, and LeVERB (Xue et al., 2025) uses a hierarchical CVAE-RL architecture for vision-language WBC but lacks support for high-dynamic motions. To overcome these issues, we propose a MoE-based oracle paired with a latent-driven diffusion student, enhancing generalization while reducing deployment cost via a retargeting-free design.

## 3  METHOD

This section presents the core components of our framework, which is depicted in Fig 2. We begin with an overview of our method in Section 3.1, providing a high-level description of the architecture and motivation. Section 3.2 details the construction of our motion generator, which leverages continuous autoregression and a causal autoencoder. Furthermore, Section 3.3 elaborates on our novel retargeting-free, latent-driven reinforcement learning architecture based on diffusion models, including its training and inference procedures. Finally, we present our causal adaptive sampling strategy in Section 3.4. Other details can be found in appendix 9.2.

### 3.1  OVERVIEW

Our work introduces a novel retargeting-free, latent-driven reinforcement learning architecture for language-guided humanoid control, which fundamentally diverges from conventional motion-tracking pipelines. As depicted in Fig. 2, our approach comprises three core components: a continuous autoregressive motion generator, a MoE-based teacher policy, and a latent-driven diffusion-based student policy. We focus on the challenge of generating diverse, physically plausible motions from high-level instructions without the need for complex kinematic retargeting.

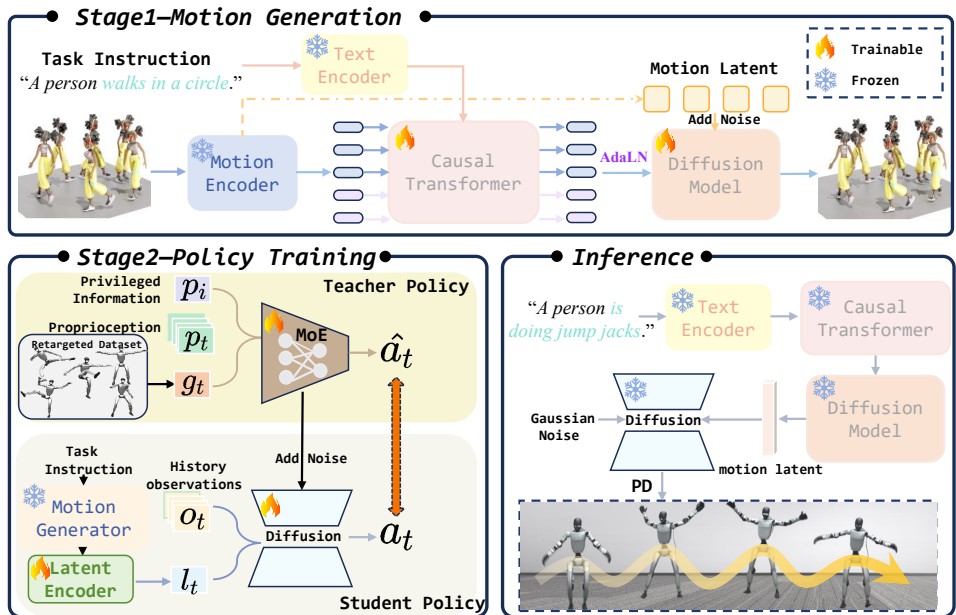

Figure 2: Overview of RoboGhost. We propose a two-stage approach: a motion latent is first generated, then a MoE-based teacher policy is trained with RL and a diffusion-based student policy is trained to denoise actions conditioned on motion latent. This latent-driven scheme bypasses the need for motion retargeting.

The process begins by feeding a textual prompt $T$ into the continuous autoregressive motion generator. Unlike prior works that decode the generated motion into an explicit kinematic sequence requiring tedious retargeting to the robot, our generator produces a compact latent motion representation $l_{ref}$. This design is pivotal, as it bypasses the error-prone retargeting step and mitigates performance degradation caused by limited generator capability. The generation process can be formulated as: $l_{ref} = G(T)$, where $G$ denotes our motion generator. This latent representation $l_{ref}$, alongside proprioceptive states and historical observations, then conditions a diffusion-based student policy $\pi_s$. The policy operates a denoising process to output actions directly executable on the physical humanoid. This latent-driven paradigm eliminates the dependency on privileged information and explicit reference motions during deployment, significantly streamlining the sim-to-real pipeline.

To efficiently train the high-level oracle teacher policy, we introduce a causal adaptive sampling strategy. It dynamically prioritizes challenging motion segments by attributing failures to their causal antecedents, thereby maximizing sample efficiency and enabling the learning of long-horizon, agile motor skills. Finally, a meticulously designed reward function ensures accurate and expressive whole-body motion tracking. Collectively, our framework achieves robust, direct-drive control from language commands, setting a new paradigm for retargeting-free humanoid locomotion.

### 3.2 CONTINUOUS AUTOREGRESSIVE MOTION GENERATOR

Given that discretized models are prone to information loss and to leverage the advantages of both masked modeling and autoregressive approaches, we adopt a causal autoencoder and continuous masked autoregressive architecture with causal attention mask, which effectively captures temporal dependencies between tokens and produces rich contextual conditions for the subsequent diffusion process. Specifically, we first randomly mask motion tokens following the practice in language models (Devlin et al., 2019), obtaining a set of masked tokens. The temporal masking strategy follows the same mask ratio scheduling as (Chang et al., 2022), defined by the function:

$$\gamma(\tau) = \cos\left(\frac{\pi\tau}{2}\right), \tag{1}$$

where $\tau \in [0, 1]$. During training, $\tau$ is uniformly sampled from $\mathbf{U}(0, 1)$, yielding a mask ratio $\gamma(\tau)$. Accordingly, $\gamma(\tau) \times N$ tokens are randomly selected for masking.

Unlike previous masked autoregressive methods (Li et al., 2024a; Meng et al., 2024; Li et al., 2023a), our approach does not involve random token shuffling or batch-token prediction. Moreover, to mitigate the limitation in model expressiveness caused by low-rank matrix approximations during training, we replace bidirectional attention masks with causal attention masking. And for the input text prompt $T$, we use the LaMP (Li et al., 2024d) text transformer to extract textual features. This model captures linguistic nuances and semantic structures effectively, resulting in high-dimensional feature representations that provide essential guidance for motion generation.

After the transformer completes the masked token prediction task, the predicted latent representations are used to condition the diffusion model, guiding the denoising process to produce more accurate and semantically rich latent representations. This enables downstream latent-driven action generation.

### 3.3 LATENT-DRIVEN DIFFUSION POLICY FOR RETARGETING-FREE DEPLOYMENT

#### 3.3.1 MoE-BASED TEACHER POLICY

The core challenge in text-to-locomotion is the inherent open-endedness of language. Therefore, generalization capability is the key enabler for policies to respond successfully to novel prompts and achieve true deployment flexibility. First, we train an oracle teacher policy using PPO (Schulman et al., 2017) with privileged simulator-state information. To learn a policy $\pi_t$ that generalizes across diverse motion inputs, we first train an initial policy $\pi_0$ on a curated motion dataset $\mathcal{D}_0$ of high diversity. We then evaluate the tracking accuracy of $\pi_0$ for each motion sequence $s \in \mathcal{D}_0$, focusing on the lower body through an error metric $e(s) = \alpha \cdot E_{\text{key}}(s) + \beta \cdot E_{\text{dof}}(s)$, where $E_{\text{key}}(s)$ denotes the mean key-body position error and $E_{\text{dof}}(s)$ the mean joint angle tracking error of the lower body. Motion sequences with $e(s) > 0.6$ are filtered out, and the remaining data $\mathcal{D}$ are used to train a general teacher policy.

The teacher policy $\pi_t$ is trained as an oracle in simulation using PPO, leveraging privileged information unavailable in the real world—including ground-truth root velocity, global joint positions, physical properties (e.g., friction coefficients, motor strength), proprioceptive state, and reference motion. The policy outputs target joint positions $\hat{a}_t \in \mathbb{R}^{23}$ for proportional derivative (PD) control, maximizing cumulative rewards to achieve accurate motion tracking and robust behavior, resulting in a high-performance expert policy trained solely on simulated data. Finally, we seek:

$$\pi_t = \arg\max_\pi \mathbb{E}_{s \in \mathcal{D}} \left[\text{Performance}(\pi, s)\right] \tag{2}$$

To enhance the expressive power and generalization capability of the model, we incorporate a Mixture of Experts (MoE) module into the training of the teacher policy. The policy network consists of a set of expert networks and a gating network. The expert networks take as input both the robot state observations and reference motion, and output the final action $a_t$. The gating network receives the same input observations and produces a probability distribution over all experts. The final action is computed as a weighted combination of actions sampled from each expert's output distribution: $a = \sum_{i=1}^{n} p_i \cdot a_i$, where $p_i$ denotes the probability assigned to the $i$-th expert by the gating network, and $a_i$ represents the output of the $i$-th expert policy. This architecture enhances the policy's generalization capacity and obtains actions that accurately track reference motions, thereby providing more precise supervised signals for the student policy.

#### 3.3.2 DIFFUSION-BASED STUDENT POLICY

Unlike prior approaches where the student policy $\pi_s$ distills knowledge from the teacher using reference motion, we propose a novel latent-driven student policy that takes motion latent representations as input. In addition to the observation history, it incorporates a latent representation $l_{ref}$ from the motion generator as input. This implicit design enables the policy to operate during inference without retargeted explicit reference motion, thereby streamlining deployment, reducing performance degradation caused by limitations of motion generator capability and retargeting.

Following a DAgger-like (Ross et al., 2011) approach, we roll out the student policy in simulation and query the teacher for optimal actions $\hat{a}_t$ at visited states. During training, we progressively inject Gaussian noise $\epsilon_t$ into the teacher's actions and use the first-stage pretrained motion generator to obtain a latent representation with textual descriptions. The forward process can be modeled as a

Markov noising process using:

$$q(x_t|x_{t-1}) = \mathcal{N}(x_t; \sqrt{1 - \alpha_t} \cdot x_{t-1}, \alpha_t \mathbf{I}) \tag{3}$$

where the constant $\alpha_t \in (0, 1)$ is a hyper-parameter for sampling. Here, we denote the denoiser as $\epsilon_\theta$, use $\{x_t\}_{t=0}^T$ to denote the noising sequence, and $x_{t-1} = \epsilon_\theta(x_t, t)$ for the t-step denoising.

While the motion generator can produce motions lacking physical realism, our method does not blindly follow its output. Instead, a trainable latent encoder intelligently conditions the policy, translating raw proposals into actionable and stable commands. This enables the policy to generate diverse and robust actions even from imperfect guidance. This representation, along with proprioceptive states and historical observations, serves as conditioning to guide the denoising process. For tractability, we adopt an $x_0$-prediction strategy and supervise the student policy by minimizing the mean squared error loss $\mathcal{L} = \|a - \hat{a}_t\|_2^2$, where $a = \frac{x_t - \sqrt{1-\bar{\alpha}_t} \cdot \epsilon_\theta(x_t, t)}{\sqrt{\bar{\alpha}_t}}$. The process iterates until convergence, yielding a policy that requires neither privileged knowledge nor explicit reference motion, and is suitable for real-world deployment.

### 3.3.3 INFERENCE PIPELINE

To ensure motion fluency and smoothness, it is essential to minimize the time required for the denoising process. We therefore adopt DDIM sampling (Song et al., 2020) and an MLP-based diffusion model to generate actions for deployment. The reverse process can be formulated as:

$$x_{t-1} = \sqrt{\alpha_{t-1}} \left( \frac{x_t - \sqrt{1 - \alpha_t} \cdot \epsilon_\theta(x_t, t)}{\sqrt{\alpha_t}} \right) + \sqrt{1 - \alpha_{t-1}} \cdot \epsilon_\theta(x_t, t) \tag{4}$$

Our framework is entirely retargeting-free and latent-driven. During inference, the textual description is first input into a motion generator and obtains a latent motion representation $l_{ref}$. Crucially, we bypass the decoding of this latent into explicit motion sequences, thus eliminating the need for retargeting to the robot. We sample a random noise as input to the student policy and condition the diffusion model via AdaLN (Huang & Belongie, 2017) on the motion latent, proprioceptive states $p_o$ and historical observations $o_{t-H:t}$, producing a clean action $a = \epsilon_\theta(\epsilon | l_{ref}, p_o, o_{t-H:t})$, that is directly executable on the physical robot. This streamlined pipeline not only reduces complexity but also mitigates issues such as low-quality motion generation due to limited generator capability, retargeting-induced errors, and insufficient action diversity.

## 3.4 CAUSAL ADAPTIVE SAMPLING

RoboGhost aims at tracking a reference motion more expressively in the whole body. To this end, we employ a novel sampling method to facilitate the teacher policy in mastering more challenging and longer motion sequences. Details about various reward functions, curriculum learning, and domain randomizations can be found in appendix 9.2.

Training long-horizon motor skills faces a fundamental challenge: motion segments exhibit heterogeneous difficulty levels. Conventional approaches typically employ uniform sampling across trajectories (He et al., 2025a; Truong et al., 2024), which leads to oversampling of trivial segments and under-sampling of challenging ones, resulting in high reward variance and suboptimal sample efficiency. To address this, we propose a causality-aware adaptive sampling mechanism. The motion sequence is divided into $K$ equal-length intervals, and the sampling probability for each interval is dynamically adjusted based on empirical failure statistics. Let $k_t$ denote the interval in which a rollout terminates due to failure. We hypothesize that the root cause often arises $s$ steps prior to $k_t$—such as a misstep or collision—that propagates into eventual failure at $k_t$. To enable corrective learning, we increase the sampling likelihood of these antecedent intervals.

Motivated by the temporal structure of failures—typically preceded by suboptimal actions—we apply an exponential decay kernel $\alpha(u) = \gamma^u$ ($\gamma \in (0, 1)$) to assign higher weights to time steps leading up to termination. The base sampling probability for interval $s$ is defined as:

$$\begin{cases} \Delta p_i = \alpha(t - i) \cdot p, & i \in [t - s, t], \\ \Delta p_i = 0, & i \notin [t - s, t] \end{cases} \tag{5}$$

where $K$ controls the backward horizon for causal attribution. Subsequently, we update and normalize the sampling probabilities across all intervals. Let $p_i$ denote the base probability for interval $i$. The updated probability is given by: $p'_i \leftarrow p_i + \Delta p_i$, where $\Delta p_i$ denotes the increment derived from failure attribution. The distribution is then renormalized to ensure $\sum_i p'_1 = 1$. Finally, the initial interval is sampled from the multinomial distribution $\text{Multinomial}(p'_1, \ldots, p'_K)$, enabling selective focus on high-difficulty segments. Since each interval consists of multiple frames, after sampling the restart interval, we further select the exact restart frame uniformly within that interval.

## 4 EXPERIMENT

We present a rigorous empirical evaluation of our proposed method in this section, structured to systematically address three core research questions central to its design and efficacy. We organize our analysis around the following three research questions:

- **Q1: Advantages of Latent-driven Pipeline.** To what extent does a retargeting-free, latent-driven approach improve efficiency, robustness, and generalization in motion generation, and how do these advantages manifest in real-world deployment scenarios?

- **Q2: Generalization Gains via Diffusion Policy.** Does replacing MLP-based policies with diffusion-based policies enhance generalization to unseen instructions or environments?

- **Q3: Better Motion Generator.** Does the continuous autoregressive architecture yield more semantically and kinematically precise latent embeddings than standard alternatives? Further, how do architectural variants of the diffusion model (e.g., MLP and DiT (Peebles & Xie, 2023)) affect motion generation quality?

- Q4: Greater Robustness Policy. What advantages does the diffusion policy offer over the MLP policy in the presence of noise interference, and will it exhibit greater robustness?

### 4.1 EXPERIMENTS SETTINGS

**Datasets and Metrics** We evaluate on two MotionMillion (Fan et al., 2025) subsets: HumanML and Kungfu. For policy training, we use only index-0 sequences per motion name (e.g., "00000_0.npy"), excluding mirrored variants to reduce redundancy. Motions with non-planar terrains or infeasible contacts are filtered to ensure flat-ground consistency. The curated set is split 8:2 (train:test) with stratified category balance. For motion generator training, we leverage the full subsets. Evaluation metrics include motion generation and motion tracking. Generation follows Guo et al. (2022b) with retrieval precision R@1, 2, 3, MM Dist, FID, and Diversity. Tracking is assessed in physics simulators, consistent with OmniH2O (He et al., 2024) and PHC (Luo et al., 2023), using success rate (primary), mean per-joint error ($E_{mpjpe}$), and mean per-keypoint error ($E_{mpkpe}$). Success rate reflects both accuracy and stability. Further details are in appendix 9.3.

### 4.2 EVALUATION OF MOTION GENERATION

To validate the effectiveness of our continuous autoregressive motion generation model, we train and evaluate it on two distinct skeleton representations: the 263-dim HumanML3D (Guo et al., 2022a) and the 272-dim Humanml and Kungfu subsets of MotionMillion. We benchmark against text-to-motion methods, including diffusion- and transformer-based approaches. Motivated by SiT (Ma et al., 2024) and MARDM (Meng et al., 2024), we reformulate the training objective of the motion generator from noise prediction to velocity prediction, leading to improved motion quality and enhanced dynamic consistency in generated sequences. As summarized in Table 1, our model achieves competitive performance across both formats, demonstrating robustness to representation variation.

### 4.3 EVALUATION OF MOTION TRACKING POLICY

To further validate the efficacy of our motion tracking policy, we conduct evaluations on the HumanML and Kungfu subsets of MotionMillion, measuring $E_{mpjpe}$ and $E_{mpkpe}$ under physics-based simulation in both IsaacGym and MuJoCo. The pipeline operates as follows: textual descriptions are first input into the motion generator to produce a latent motion representation, which is subsequently input by the student policy for action execution. As summarized in Table 2, our method achieves high

| Methods | R Precision↑ | | | FID↓ | MM-Dist↓ | Diversity→ |
|---|---|---|---|---|---|---|
| | Top 1 | Top 2 | Top 3 | | | |
| HumanML3D | | | | | | |
| Ground Truth | 0.702 | 0.864 | 0.914 | 0.002 | 15.151 | 27.492 |
| MDM (Tevet et al., 2023) | 0.523 | 0.692 | 0.764 | 23.454 | 17.423 | 26.325 |
| MLD (Chen et al., 2023) | 0.546 | 0.730 | 0.792 | 18.236 | 16.638 | 26.352 |
| T2M-GPT (Zhang et al., 2023a) | 0.606 | 0.774 | 0.838 | 12.475 | 16.812 | 27.275 |
| MotionGPT (Jiang et al., 2023) | 0.456 | 0.598 | 0.628 | 14.375 | 16.892 | 27.114 |
| MoMask (Guo et al., 2023) | 0.621 | 0.784 | 0.846 | 12.232 | 16.138 | 27.127 |
| AttT2M (Zhong et al., 2023) | 0.592 | 0.765 | 0.834 | 15.428 | 15.726 | 26.674 |
| MotionStreamer (Xiao et al., 2025) | 0.631 | 0.802 | 0.859 | 11.790 | 16.081 | 27.284 |
| Ours-DDPM | 0.639 | 0.808 | 0.867 | 11.706 | 15.772 | 27.230 |
| Ours-SiT | 0.641 | 0.812 | 0.870 | 11.743 | 15.663 | 27.307 |
| HumanML (MotionMillion) | | | | | | |
| Ground Truth | 0.714 | 0.876 | 0.920 | 0.002 | 14.984 | 26.571 |
| Ours-DDPM | 0.644 | 0.819 | 0.873 | 11.724 | 15.870 | 26.395 |
| Ours-SiT | 0.646 | 0.818 | 0.872 | 11.716 | 15.603 | 26.471 |

Table 1: Quantitative results of text-to-motion generation on the HumanML3D dataset and HumanML subset. → denotes that if the value is closer to the ground truth, the metric is better.

| Method | IsaacGym | | | MuJoCo | | |
|---|---|---|---|---|---|---|
| | Succ ↑ | $E_{mpjpe}$ ↓ | $E_{mpkpe}$ ↓ | Succ ↑ | $E_{mpjpe}$ ↓ | $E_{mpkpe}$ ↓ |
| HumanML (MotionMillion) | | | | | | |
| Baseline | 0.92 | 0.23 | 0.19 | 0.64 | 0.34 | 0.31 |
| Ours-DDPM | 0.97 | 0.12 | 0.09 | 0.74 | 0.24 | 0.20 |
| Ours-SiT | 0.98 | 0.14 | 0.08 | 0.72 | 0.26 | 0.23 |
| Kungfu (MotionMillion) | | | | | | |
| Baseline | 0.66 | 0.43 | 0.37 | 0.51 | 0.58 | 0.52 |
| Ours-DDPM | 0.72 | 0.34 | 0.31 | 0.57 | 0.54 | 0.50 |
| Ours-SiT | 0.71 | 0.36 | 0.32 | 0.55 | 0.53 | 0.48 |

Table 2: Motion tracking performance comparison in simulation on the HumanML and Kungfu test sets.

success rates on HumanML, alongside low joint and keypoint errors, indicating robust alignment between generated motion semantics and physically executable trajectories. Here, Baseline refers to the conventional, explicit framework where both the teacher and student policies use MLP backbones.

## 4.4 QUALITATIVE EVALUATION

We present a qualitative evaluation of the motion tracking policy across three deployment stages: simulation (IsaacGym), cross-simulator transfer (MuJoCo), and real-world execution on the Unitree G1 humanoid. Fig 3 visualizes representative tracking sequences, highlighting the policy's ability to preserve motion semantics, maintain balance under dynamic transitions, and generalize across physics engines and hardware. In particular, real-robot deployments demonstrate that our latent-driven, retargeting-free framework enables smooth, temporally coherent motion execution without manual tuning, validating its readiness for practical embodied applications.

## 4.5 ABLATION STUDIES

To systematically answer the three research questions posed at the outset of this section, we conduct various ablation studies.

IsaacGym                                    MuJoCo

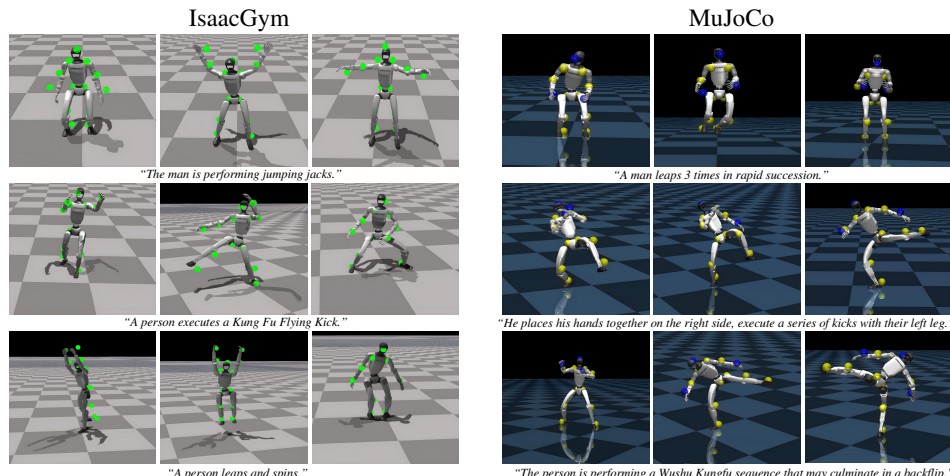

Figure 3: Qualitative results in the IsaacGym and MuJoCo.

| Method | Time (s) | Succ ↑ | $E_{mpjpe}$ ↓ | $E_{mpkpe}$ ↓ |
|---|---|---|---|---|
| PHC-100 | 1.63 | 0.81 | 0.45 | 0.40 |
| PHC-500 | 6.09 | 0.88 | 0.31 | 0.25 |
| PHC-800 | 9.87 | 0.91 | 0.25 | 0.21 |
| PHC-1000 | 11.89 | 0.93 | 0.21 | 0.17 |

Table 3: Average inference time and tracking performance on different retargeting methods.

To answer **Q1 (Advantages of Retargeting-free Pipeline)**, we evaluate the conventional pipeline: text prompts are fed to the motion generator, the output latent is decoded into explicit motion, which is then retargeted to the G1 robot and executed by the policy. As shown in Table 4, this approach underperforms due to its multi-stage complexity and error accumulation, and incurs higher real-world deployment time cost. This confirms the efficiency and performance advantage of our retargeting-free, latent-driven framework.

Furthermore, we investigate the time consumption and corresponding tracking performance of different PHC retargeting iteration steps under the explicitly-driven framework. As shown in Table 3, reducing the number of PHC iteration steps accelerates the retargeting process, but significantly degrades the tracking performance.

To answer **Q2 (Generalization Gains via Diffusion Policy)**, we conduct an ablation by replacing the diffusion policy with an MLP-based policy that concatenates latent and other observations to predict actions. Diffusion policies, by design, better capture various action distributions, enabling more robust adaptation to diverse or imperfect latents. As shown in Table 5, the diffusion policy significantly outperforms its MLP counterpart.

To further evaluate generalization, we test both policies on 10 randomly sampled motions from unseen MotionMillion subsets (fitness, perform, 100style, haa). Although the motion generator was not trained on these subsets, resulting in suboptimal latents, the diffusion-based policy still achieves substantially better tracking and robustness than MLP, as evidenced in Table 5. This highlights the humanoid diffusion policy's superiority.

To address **Q3 (Better Motion Generator)**, we conduct an ablation study on the diffusion backbone in our framework, comparing a 16-layer MLP and a 4-layer DiT under identical settings. As shown in Table 6, DiT offers slight gains in generation metrics but no measurable improvement in tracking success or joint accuracy, while incurring higher latency due to larger model size. Balancing effectiveness and efficiency, we therefore adopt the 16-layer MLP as our default backbone.

To address Q4 (Greater Robustness Policy), we present the tracking performance of different methods on the same motion in simulation in the upper part of Figure 4. When tracking highly dynamic motions, the MLP policy performs significantly worse than the diffusion policy. Furthermore, we

| Method | IsaacGym | | | MuJoCo | | | Time Cost (s) |
|---|---|---|---|---|---|---|---|
| | Succ ↑ | $E_{mpjpe}$ ↓ | $E_{mpkpe}$ ↓ | Succ ↑ | $E_{mpjpe}$ ↓ | $E_{mpkpe}$ ↓ | |
| Ours-Explicit | 0.93 | 0.21 | 0.17 | 0.66 | 0.32 | 0.27 | 17.85 |
| Ours-Implicit | 0.97 | 0.12 | 0.09 | 0.74 | 0.24 | 0.20 | 5.84 |

Table 4: Motion tracking performance comparison across different simulators on the HumanML and Kungfu test sets. Explicit version including PHC retargeting (1000 interations) and latent decode processes.

| Method | IsaacGym | | |
|---|---|---|---|
| | Succ ↑ | $E_{mpjpe}$ ↓ | $E_{mpkpe}$ ↓ |
| MLP Policy | 0.96 | 0.17 | 0.11 |
| Diffusion Policy | 0.97 | 0.12 | 0.09 |

| Method | IsaacGym | | |
|---|---|---|---|
| | Succ ↑ | $E_{mpjpe}$ ↓ | $E_{mpkpe}$ ↓ |
| MLP Policy | 0.54 | 0.48 | 0.45 |
| Diffusion Policy | 0.68 | 0.42 | 0.39 |

Table 5: Comparison of MLP-based and diffusion-based policy. The left table shows the tracking performance on HumanML subset, and the right table presents the generalization ability of two different policy architectures.

evaluate the robustness of different policies. As shown in the lower part of Figure 4, in the MuJoCo simulator, when we randomly add noise with a noise scale of 0.2 to the observations of both policies, the MLP policy maps this noise to noisy actions, ultimately causing the robot to fall. In contrast, the diffusion policy exhibits strong robustness against such noise and still achieves favorable tracking performance. We also investigate the maximum tolerable noise scale for both policies: the MLP policy can only complete the task when the noise scale is 0.12, while the diffusion policy maintains task completion at a noise scale of up to 0.33.

| Method | IsaacGym | | | HumanML3D | | Time Cost (s) ↓ |
|---|---|---|---|---|---|---|
| | Succ ↑ | $E_{mpjpe}$ ↓ | $E_{mpkpe}$ ↓ | Top 3 ↑ | FID ↓ | |
| DiT | 0.96 | 0.11 | 0.11 | 0.870 | 11.697 | 14.28 |
| MLP | 0.97 | 0.12 | 0.09 | 0.867 | 11.706 | 5.84 |

Table 6: Comparison of tracking performance across different diffusion backbones for motion generation.

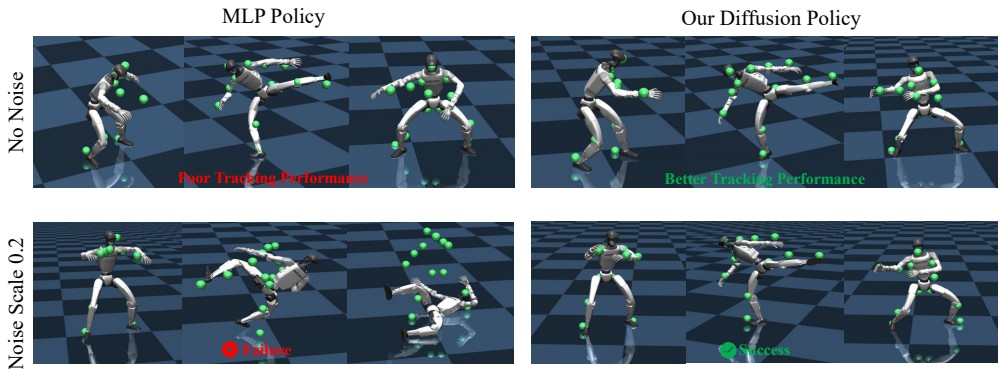

Figure 4: Comparison between our diffusion policy and MLP policy regarding their tracking performance under noise-free conditions and robustness after the introduction of noise.

## 5 CONCLUSION

In this paper, we introduce RoboGhost, a retargeting-free, latent-driven framework that bridges natural language instructions with physically feasible humanoid motion control. Our method harnesses rich semantic representations from a pretrained text-to-motion model and integrates them into a diffusion-based humanoid policy, thereby bypassing motion decoding and retargeting stages that are not only error-prone but also time-consuming. This design enables real-time, language-guided humanoid locomotion with improved adaptability. Extensive experiments show that RoboGhost reduces cumulative distortions and execution latency while preserving semantic alignment and physical realism. Beyond advancing efficiency and robustness, RoboGhost offers a new and practical pathway toward more intuitive, responsive, and deployable humanoid robots in real-world environments.

## 6 ETHICS STATEMENT

This research was conducted exclusively in simulation environments (IsaacGym, MuJoCo) and on humanoid robot hardware. It does not involve the use of any human subjects or personal identifiable data. All motion data utilized are sourced from publicly available, authorized human motion capture databases. The primary goal of our work is to advance the field of humanoid robot control, with potential applications (e.g., assistive robotics) aimed at benefiting society. We have carefully considered the fairness and safety of our algorithms and are committed to developing robust control systems to minimize the risk of unintended physical harm in real-world deployment. There are no known conflicts of interest associated with this study.

## 7 REPRODUCIBILITY STATEMENT

To ensure the reproducibility of this work, we have provided comprehensive details in the main text and appendix, including the network architecture, hyperparameter settings, and curriculum learning strategies. All motion sequence data used for training are sourced from publicly available datasets. Demo videos have been included in the supplementary material for review. Detailed descriptions of the dynamic model, reward function design, and termination conditions can be found in Section 3 and Appendix A. We encourage interested readers to consult these resources to replicate our results. Upon acceptance of the paper, the relevant code will be made publicly available.

## 8 THE USE OF LARGE LANGUAGE MODELS (LLMS)

LLMs were used in the preparation of this work solely for the purpose of grammar correction and proofreading. The LLM did not contribute to the ideation, scientific content, writing, or analysis of the research. The authors take full responsibility for the entire intellectual content of this paper.

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

## 9 APPENDIX

### APPENDIX OVERVIEW

This appendix provides additional details and results, organized as follows:

- **Section 9.1**: Detailed information of implementation, including detailed proprioceptive states, privileged information and hyper-parameters of network structure.

- **Section 9.2**: Elaboration on some details during training, including dataset details, motion filter and retargeting, simulator, domain randomization, regularization, reward functions, penalty and curriculm learning.

- **Section 9.3**: Details about evaluation, including metrics about motion generation and motion tracking.

- **Section 9.4**: Deployment details, including sim-to-sim and sim-to-real.

- **Section 9.5**: Extra qualitative experiment results and visualizations, including in the simulation and in the real-world.

- **Section 9.6**: Extra ablation experiment results and visualizations, including the effect of causal adaptive sampling (CAS), hyper-parameter $\lambda$, and the number of experts in MoE of teacher policy.

- **Section 9.7**: Pseudo codes of the algorithmic workflow and details of the diffusion student policy.

### 9.1 IMPLEMENTATION DETAILS

In this section, we describe the state representation used for policy training, covering proprioceptive states, privileged information, and hyper-parameters of our network. The proprioceptive state components for both the teacher and student policies are summarized in Table 7. A key distinction is that the student policy utilizes a longer history of observations, compensating for its lack of access to privileged information by relying on extended temporal context.

For privileged information, the teacher policy incorporates privileged information to achieve precise motion tracking. The full set of privileged state features is provided in Table 7. Previous methods receive motion target information as part of the observation in both teacher and student policy, which includes keypoint positions, desired joint (DoF) positions, and root velocity. But in our method, only the teacher policy receives these information, and the student policy only receives the latent from motion generator. A detailed target state composition is presented in Table 8. The action output corresponds to target joint positions for proportional-derivative (PD) control, with a dimensionality of 23 for the Unitree G1 humanoid platform.

RoboGhost employs a teacher-student training framework. The teacher policy, trained via standard PPO (Schulman et al., 2017), utilizes privileged information, tracking targets, and proprioceptive states. In contrast, the student policy is trained using Dagger without access to privileged information, but instead relies on an extended history of observations. Furthermore, while the teacher policy uses explicit reference motion information, the student policy replaces this with a latent representation from a motion generator, resulting in a retargeting-free and latent-driven approach. For the teacher policy, inputs are concatenated and processed by a Mixture-of-Experts (MoE) network for policy learning. The student policy feeds its concatenated inputs as conditions to a diffusion model with an MLP backbone. Furthermore, we employ AdaLN to inject conditional information throughout the denoising process within the diffusion model. A final MLP layer is appended to the backbone network, which projects the output to a 23-dimensional action space while additionally incorporating the conditional signals. Detailed hyperparameters for both policies are provided in Table 9.

### 9.2 TRAINING DETAILS

**Dataset Details** Our motion generator is pretrained on the MotionMillion dataset (Fan et al., 2025). Given the substantial scale of this dataset, we restrict pretraining to the humanml and kungfu subsets from the MotionUnion branch, comprising a total of 50,378 motion sequences. During policy training,

**Proprioceptive States**

| State Component | Dim. |
|---|---|
| DoF position | 23 |
| DoF velocity | 23 |
| Last Action | 23 |
| Root Angular Velocity | 3 |
| Roll | 1 |
| Pitch | 1 |
| Yaw | 1 |
| Total dim | $75 \times 10$ |

**Privileged Information**

| | |
|---|---|
| DoF Difference | 23 |
| Keybody Difference | 36 |
| Root velocity | 3 |
| Total dim | 62 |

Table 7: Proprioceptive states and privileged information.

**Teacher Policy**

| State Component | Dim. |
|---|---|
| DoF position | 23 |
| Keypoint position | 36 |
| Root Velocity | 3 |
| Root Angular Velocity | 3 |
| Roll | 1 |
| Pitch | 1 |
| Yaw | 1 |
| Height | 1 |
| Total dim | 69 |

**Student Policy**

| | |
|---|---|
| Latent | 64 |
| Total dim | 64 |

Table 8: Reference information in the teacher and student policies.

| Hyperparameter | Value |
|---|---|
| Optimizer | AdamW |
| $\beta_1, \beta_2$ | 0.9, 0.999 |
| Learning Rate | $1 \times 10^{-4}$ |
| Batch Size | 4096 |
| **Teacher Policy** | |
| Discount factor ($\gamma$) | 0.99 |
| Clip Parameter | 0.2 |
| Entropy Coefficient | 0.005 |
| Max Gradient Norm | 1 |
| Learning Epochs | 5 |
| Mini Batches | 4 |
| Value Loss Coefficient | 1 |
| Value MLP Size | [512, 256, 128] |
| Actor MLP Size | [512, 256, 128] |
| Experts | 5 |
| **Student Policy** | |
| MLP Layers | 4 |
| MLP Size | [256, 256, 256] |

Table 9: Hyperparameters for teacher and student policy training.

we again draw data from the humanml and kungfu subsets. However, due to the presence of extensive duplicate and mirrored sequences, we perform a data cleaning process to remove redundancies and non-flat-ground motions. Using this filtered dataset, we first train an initial policy, denoted as $\pi_0$. This policy is then used to further filter the dataset by excluding sequences with a tracking error exceeding 0.6. After this refinement, the humanml subset contains 3,261 motion sequences, and the kungfu subset contains 200. And due to the significant domain gap and difference in complexity between these two subsets, we opt to train two separate policies, each specialized on one subset.

**Motion Filter and Retargeting**   Following Xie et al. (2025) and Tripathi et al. (2023), we compute the ground-projected distance between the center of mass (CoM) and center of pressure (CoP) for each frame and apply a stability threshold. Let $\bar{\mathbf{p}}_t^{\text{CoM}} = (p_{t,x}^{\text{CoM}}, p_{t,y}^{\text{CoM}})$ and $\bar{\mathbf{p}}_t^{\text{CoP}} = (p_{t,x}^{\text{CoP}}, p_{t,y}^{\text{CoP}})$ denote the 2D ground projections of CoM and CoP at frame $t$, respectively. Let $\Delta d_t = \|\bar{\mathbf{p}}_t^{\text{CoM}} - \bar{\mathbf{p}}_t^{\text{CoP}}\|_2$.

We define the stability criterion of a frame as $\Delta d_t < \epsilon_{\text{stab}}$. A motion sequence will not be filtered if its first and last frames are stable, and the longest consecutive unstable segment is shorter than 100.

**Simulator**    Following established protocols in motion tracking policy research (Ji et al., 2024; He et al., 2025a; Ji et al., 2025; Team et al., 2025), we also adopt a three-stage evaluation pipeline: (1) large-scale reinforcement learning training in IsaacGym; (2) zero-shot transfer to MuJoCo to assess cross-simulator generalization; and (3) physical deployment on the Unitree G1 humanoid platform to validate the performance in real-world.

**Reference State Initialization**    Task initialization is a critical factor in reinforcement learning (RL) training. We observe that naïvely initializing episodes from the beginning of the reference motion frequently leads to policy failure, especially for difficult motions. This can cause the environment to overfit to easier frames and fail to learn the most challenging segments of the motion.

To mitigate this issue, we employ the Reference State Initialization (RSI) framework (Peng et al., 2018). Concretely, we sample time-phase variables uniformly over $[0, 1]$, randomizing the starting point within the reference motion that the policy must track. The robot's state—including root position, orientation, linear and angular velocities, as well as joint positions and velocities—is then initialized to the values from the reference motion at the sampled phase. This approach substantially enhances motion tracking performance, particularly for highly-dynamic whole-body motions, by enabling the policy to learn various segments of the movement in parallel rather than being limited to a strictly sequential learning process.

**Domain Randomization and Regularization**    To improve the robustness and generalization of the pretrained policy, we utilize the domain randomization techniques and regularization items, which are listed in Table 10.

| Domain Randomization | |
|---|---|
| **Term** | **Value** |
| Friction | $\mathcal{U}(0.5, 2.2)$ |
| P Gain | $\mathcal{U}(0.75, 1.25) \times$ default |
| Control delay | $\mathcal{U}(20, 40)\,\text{ms}$ |
| **External Perturbation** | |
| Push robot | interval $= 8\,\text{s}$, $v_{xy} = 0.5\,\text{m/s}$ |
| **Regularization** | |
| **Term Expression** | **Weight** |
| DoF position limits $\mathbf{1}(d_t \notin [q_{\min}, q_{\max}])$ | $-10$ |
| DoF acceleration $\|\ddot{d}_t\|_2^2$ | $-3 \times 10^{-7}$ |
| DoF error $\|d_t - d_0\|_2^2$ | $-0.1$ |
| Action rate $\|a_t - a_{t-1}\|_2^2$ | $-0.5$ |
| Feet air time $T_{\text{air}} - 0.5$ | $10$ |
| Feet contact force $\|F_{\text{feet}}\|_2^2$ | $-0.003$ |
| Stumble $\mathbf{1}(F_{\text{feet}}^x > 5 \times F_{\text{feet}}^z)$ | $-2$ |
| Waist roll pitch error $\|p_t^{\text{wrp}} - p_0^{\text{wrp}}\|_2^2$ | $-0.5$ |
| Ankle Action $\|a_t^{\text{ankle}}\|_2^2$ | $-0.3$ |

Table 10: Domain randomization and regularization parameters.

**Motion Tracking Rewards**    We define the reward function as the sum of penalty, regularization, and task rewards, which is meticulously designed to improve both the performance and motion realism of the humanoid robot. It consists of terms that incentivize accurate tracking of root velocity, direction, and orientation, as well as precise keypoints and joints position tracking. Inspired by ASAP (He et al., 2025a), we additionally incorporate regularization terms to enhance stability and sim-to-real transferability, including torques, action rate, feet orientation, feet heading, and slippage. In terms of

penalty, we adopt penalty terms for DoF position limits, torque limits, and termination. The detailed reward functions are summarized in Table 11, more terms can be seen in the supplementary material.

| Task Reward | | | |
|---|---|---|---|
| Root velocity | 10.0 | Root velocity direction | 6.0 |
| Root angular velocity | 1.0 | Keypoint position | 10.0 |
| Feet position | 12.0 | DoF position | 6.0 |
| DoF velocity | 6.0 | | |
| Penalty | | | |
| DoF position limits | $-10.0$ | Torque limits | $-5.0$ |
| Termination | $-200.0$ | | |

Table 11: Reward terms for pretraining

**Curriculum Learning** Training policies to track agile motions in simulation is challenging, as certain dynamic behaviors are too difficult for the policy to learn effectively from the outset. To mitigate this issue, we utilize a termination curriculum (He et al., 2025a) that progressively tightens the motion tracking error tolerance during training, thereby guiding the policy to gradually improve its tracking fidelity. Initially, we set a lenient termination threshold of 1.5m—episodes are terminated if the robot deviates from the reference motion by more than this margin. As training progresses, the threshold is annealed down gradually, incrementally increasing the precision required for successful tracking. This curriculum enables the policy to first acquire basic balancing skills before refining them under stricter tracking constraints, ultimately facilitating the successful execution of highly dynamic behaviors.

### 9.3 EVALUATION DETAILS

**Motion Generation Metrics** Following (Guo et al., 2022c; Li et al., 2024d;b), we evaluate our approach using established text-to-motion metrics: retrieval accuracy (R@1, R@2, R@3), Multimodal Distance (MMDist), and Fréchet Inception Distance (FID).

- Retrieval Accuracy (R-Precision): These metrics measure the relevance of generated motions to text descriptions in a retrieval setup. R@1 denotes the fraction of text queries for which the correct motion is retrieved as the top match, reflecting the model's precision in identifying the most relevant motion. R@2 and R@3 extend this notion, indicating recall within the top two and three retrieved motions, respectively.

- Multimodal Distance (MMDist): This quantifies the average feature-space distance between generated motions and their corresponding text embeddings, typically extracted via a pretrained retrieval model. Smaller MMDist values indicate stronger semantic alignment between text descriptions and motion outputs.

- Fréchet Inception Distance (FID): FID assesses the quality and realism of generated motions by comparing their feature distribution to that of real motion data using a pretrained feature encoder (e.g., an Inception-style network). It computes the Fréchet distance between multivariate Gaussian distributions fitted to real and generated motion features. Lower FID scores reflect better distributional similarity and higher perceptual quality.

- Diversity: Diversity quantifies the variance of motion sequences within the dataset. It is computed by randomly sampling $N_{\text{diver}} = 300$ pairs of motions, denoted as $(f_{i,1}, f_{i,2})$ for each pair $i$. The metric is calculated as $\frac{1}{N_{\text{diver}}} \sum_{i=1}^{N_{\text{diver}}} \|f_{i,1} - f_{i,2}\|$

**Motion Tracking Metrics** For motion tracking evaluation, we adopt metrics commonly used in prior work (Ji et al., 2024): Success Rate (Succ), Mean Per Joint Position Error ($E_{mpjpe}$), and Mean Per Keybody Position Error ($E_{mpkpe}$).

- Success Rate (Succ): This metric evaluates whether the humanoid robot successfully follows the reference motion without falling. A trial is considered a failure if the average trajectory

deviation exceeds 0.5 meters at any point, or if the root pitch angle passes a predefined threshold.

- Mean Per Joint Position Error ($E_{mpjpe}$ in rad): $E_{mpjpe}$ measures joint-level tracking accuracy by computing the average error in degrees of freedom (DoF) rotations between the reference and generated motion.

- Mean Per Keybody Position Error ($E_{mpkpe}$ in m): $E_{mpkpe}$ assesses keypoint tracking performance based on the average positional discrepancy between reference and generated keypoint trajectories.

### 9.4 DEPLOYMENT DETAILS

**Sim-to-Sim** As demonstrated in Humanoid-Gym (Gu et al., 2024), MuJoCo exhibits more realistic dynamics compared to Isaac Gym. Following established practices in motion tracking policy research (Ji et al., 2024), we perform reinforcement learning training in Isaac Gym to leverage its computational efficiency. To assess policy robustness and generalization, we then conduct zero-shot transfer to the MuJoCo simulator. Finally, we deploy the policy on a physical humanoid robot to evaluate the effectiveness of our RoboGhost framework for real-world motion tracking.

**Sim-to-Real** Our real-world experiments utilize a Unitree G1 humanoid robot, equipped with an onboard Jetson Orin NX module for computation and communication. The control policy takes motion-tracking targets as input, computes target joint positions, and issues commands to the robot's low-level controller at 50Hz. Command transmission introduces a latency of 18–30ms. The low-level controller operates at 500Hz to ensure stable real-time actuation. Communication between the policy and the low-level interface is implemented using the Lightweight Communications and Marshalling (LCM) (Huang et al., 2010).

### 9.5 ADDITIONAL QUALITATIVE RESULTS

**Qualitave Results in simluation and real-world** To further validate the effectiveness of our method, we provide additional qualitative results in this section, including motion tracking visualizations in IsaacGym and MuJoCo (Fig 5) and real-world locomotion demonstrations on the physical robot (Fig 6). Moreover, we provide additional motion generation qualtative results in Fig 7. A supplementary video showcasing real-robot deployments is provided in the supplementary material.

### 9.6 ADDITIONAL ABLATION STUDIES

**Hyperparameters of Causal Adaptive Sampling** To rigorously evaluate the effectiveness of causal adaptive sampling (CAS), we conduct various ablation studies examining both its performance gains and sensitivity to hyperparameters. As shown in Table 12, causal adaptive sampling increases the sampling probability of kinematically challenging frames within a motion sequence, thereby improving training efficiency and enhancing model generalization. Furthermore, as illustrated in Fig 8a and 8b, optimal performance is achieved when the adaptation parameters are set to $\lambda = 0.8$ and $p = 0.005$.

**Number of Experts in MoE** Furthermore, we conduct an ablation study on the number of experts in the MoE architecture. As illustrated in Fig 9, the variation in the number of experts has a measurable yet limited impact on the policy's performance, with the optimal result achieved when the number of experts is set to 5.

**Tracking Performance Against Tracking Policy** To further validate the effectiveness of our policy, we conduct comparisons with other tracking policies. Since the Baseline in Table 2 of the main paper is trained and evaluated on our dataset using the model architecture and settings of Exbody2 (Ji et al., 2024), we supplement here the test results of the policy trained with the model architecture and settings of GMT (Chen et al., 2025), as shown in Table 13. The specific process is as follows: we train these policies on our dataset and evaluate their tracking performance on our test set. Since both are explicitly-driven policies, the student policy takes reference motion as input, we first generate explicit motion via our motion generator, which is then retargeted and fed to the corresponding student policy to obtain the final results.

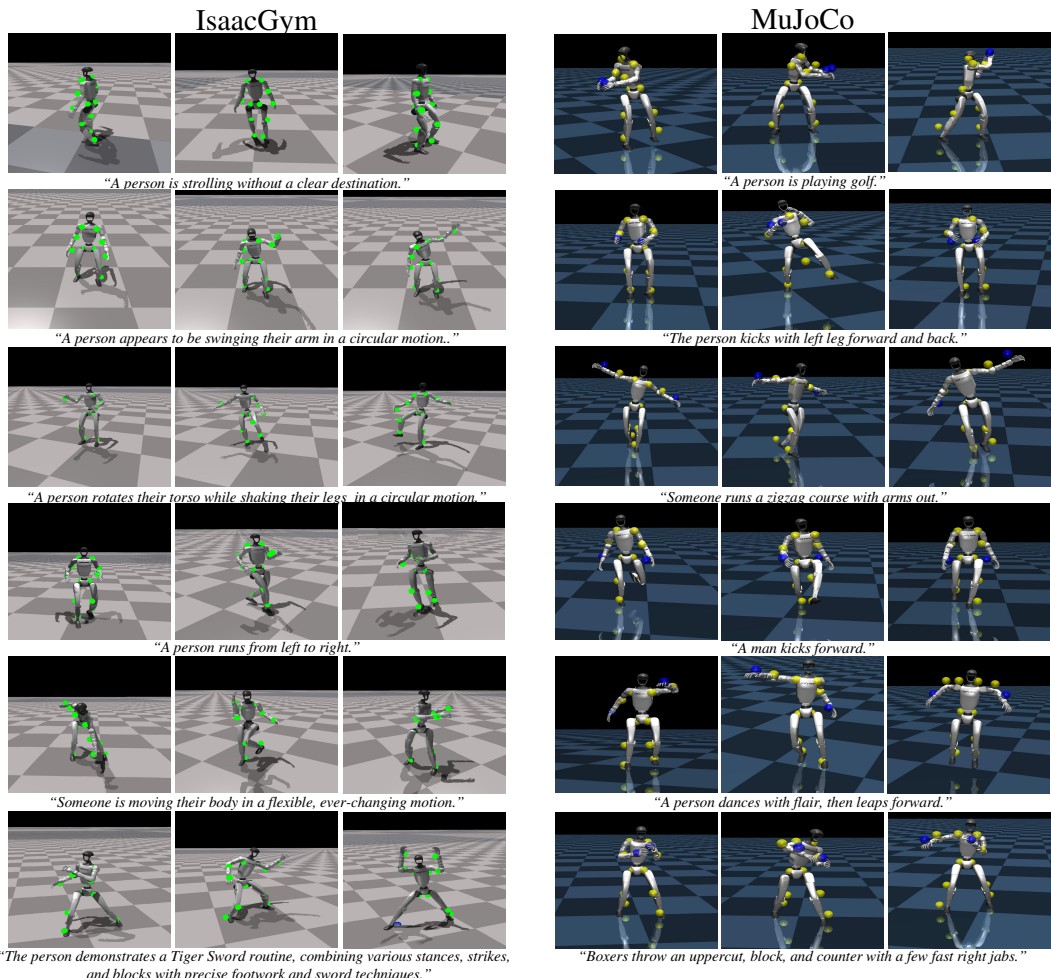

Figure 5: Qualitative results in the IsaacGym and MuJoCo.

| Method | IsaacGym | | | MuJoCo | | |
|---|---|---|---|---|---|---|
| | Succ ↑ | $E_{mpjpe}$ ↓ | $E_{mpkpe}$ ↓ | Succ ↑ | $E_{mpjpe}$ ↓ | $E_{mpkpe}$ ↓ |
| HumanML (MotionMillion) | | | | | | |
| w/o CAS | 0.92 | 0.18 | 0.14 | 0.68 | 0.32 | 0.28 |
| Ours | 0.97 | 0.12 | 0.09 | 0.74 | 0.24 | 0.20 |
| Kungfu (MotionMillion) | | | | | | |
| w/o CAS | 0.62 | 0.42 | 0.41 | 0.48 | 0.67 | 0.63 |
| Ours | 0.72 | 0.34 | 0.31 | 0.57 | 0.54 | 0.50 |

Table 12: Ablation study on causal adaptive sampling.

**Denoising Steps in Student Policy** we test the inference time and tracking performance of DDIM sampling with 2, 4, 6, 8, and 10 steps. Notably, inference time is largely independent of motion sequences, so we only report the average time per step for deploying an action. From Table 14, it can be observed that increasing the number of sampling steps yields almost no improvement in tracking performance, but significantly increases the time per step. This introduces inference latency on the real humanoid robot, thereby affecting deployment outcomes.

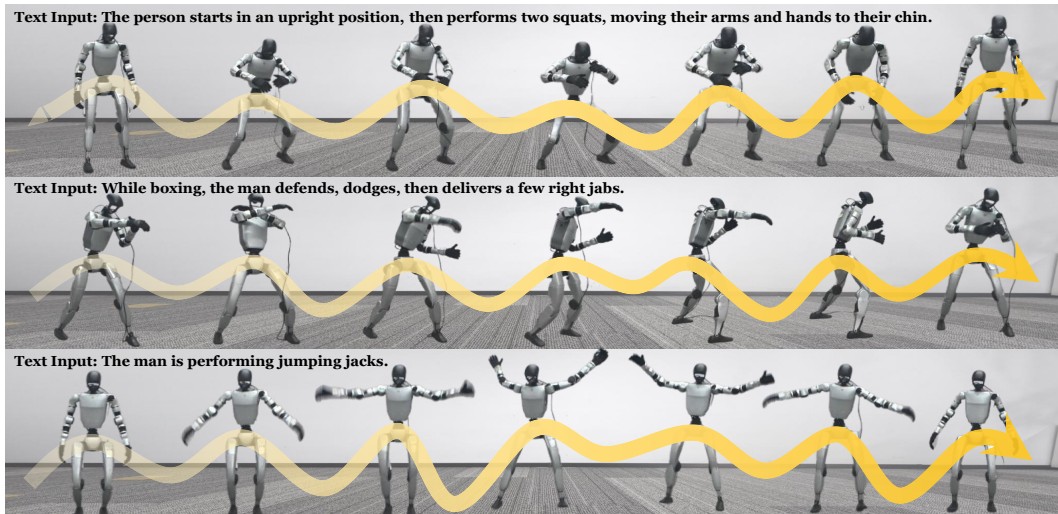

Figure 6: Qualitative results on the Unitree G1.

| Method | IsaacGym | | | MuJoCo | | |
|---|---|---|---|---|---|---|
| | Succ ↑ | $E_{mpjpe}$ ↓ | $E_{mpkpe}$ ↓ | Succ ↑ | $E_{mpjpe}$ ↓ | $E_{mpkpe}$ ↓ |
| HumanML (MotionMillion) | | | | | | |
| Exbody2 (Ji et al., 2024) | 0.92 | 0.23 | 0.19 | 0.64 | 0.34 | 0.31 |
| GMT (Chen et al., 2025) | 0.95 | 0.15 | 0.12 | 0.73 | 0.25 | 0.22 |
| Ours | 0.97 | 0.12 | 0.09 | 0.74 | 0.24 | 0.20 |
| Kungfu (MotionMillion) | | | | | | |
| Exbody2 (Ji et al., 2024) | 0.66 | 0.43 | 0.37 | 0.51 | 0.58 | 0.52 |
| GMT (Chen et al., 2025) | 0.70 | 0.36 | 0.33 | 0.54 | 0.56 | 0.52 |
| Ours | 0.72 | 0.34 | 0.31 | 0.57 | 0.54 | 0.50 |

Table 13: Motion tracking performance comparison against other tracking policies in simulation on the HumanML and Kungfu test sets.

**Noise Scale in Student Policy**  We test the inference time and tracking performance of DDIM sampling with 2, 4, 6, 8, and 10 steps. Notably, inference time is largely independent of motion sequences, so we only report the average time per step for deploying an action. From Table 15, it can be observed that increasing the number of sampling steps yields almost no improvement in tracking performance, but significantly increases the time per step. This introduces inference latency on the real humanoid robot, thereby affecting deployment outcomes.

**Noise Schedule Strategies in Student Policy**  We test the inference time and tracking performance of DDIM sampling with 2, 4, 6, 8, and 10 steps. Notably, inference time is largely independent of motion sequences, so we only report the average time per step for deploying an action. From Table 16, it can be observed that increasing the number of sampling steps yields almost no improvement in tracking performance, but significantly increases the time per step. This introduces inference latency on the real humanoid robot, thereby affecting deployment outcomes. The success rate and latency curves are presented in Figure 10.

**Optimization Objective in Student Policy**  We conduct ablation experiments on the diffusion policy with respect to different supervision targets, including $x0$-prediction and $\epsilon$-prediction. As shown in Table 17, adopting $x0$-prediction as the optimization target for the diffusion policy yields significantly better tracking performance. Furthermore, we observe that higher feature dimensions lead to worse results when using $\epsilon$-prediction.

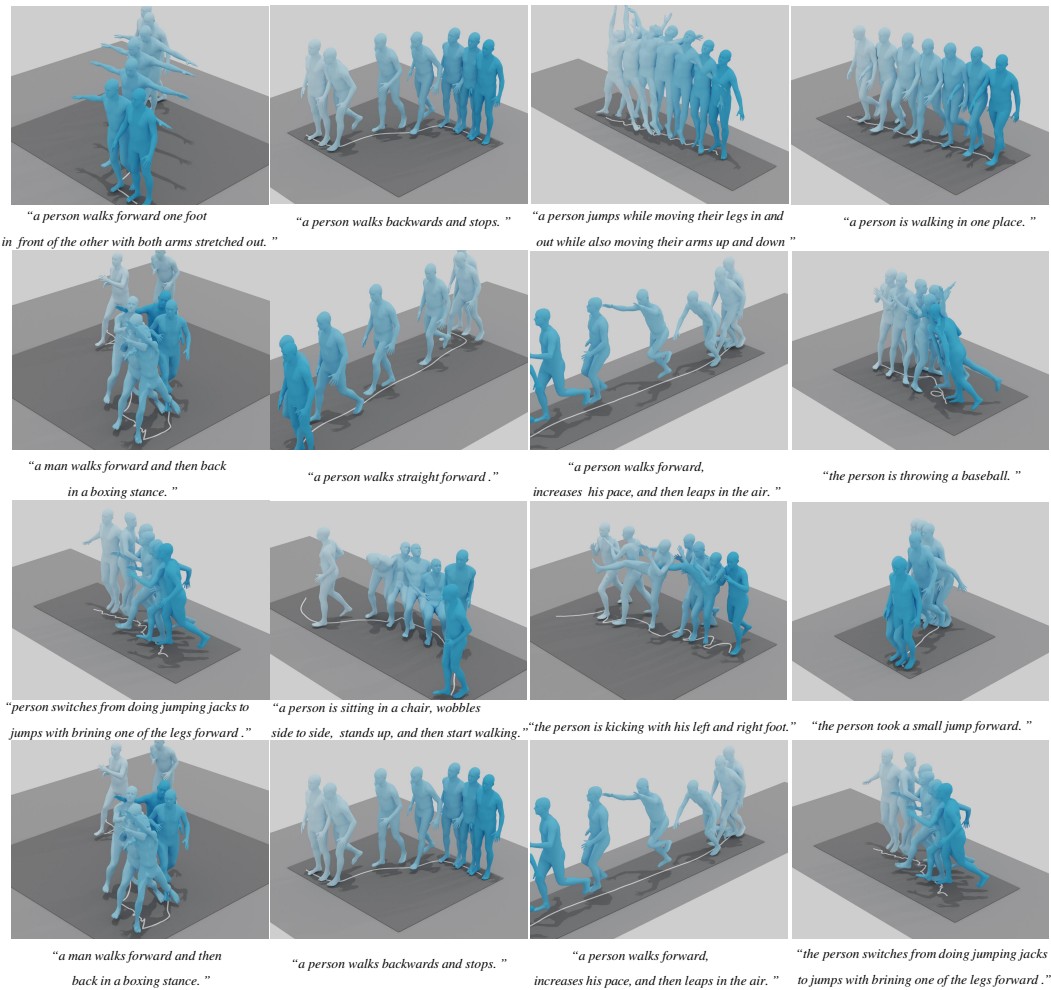

Figure 7: Qualitative results of motion generator.

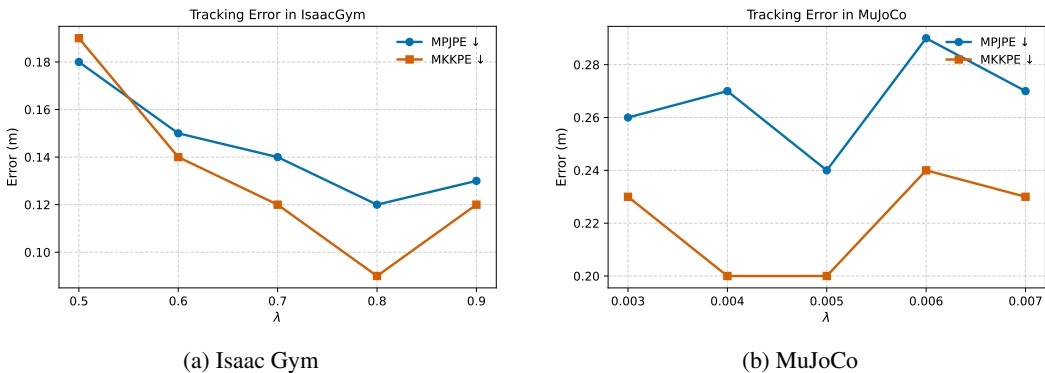

(a) Isaac Gym               (b) MuJoCo

Figure 8: The impact of different $\lambda$ values on tracking performance in IsaacGym and MuJoCo.

**Retargeting Methods and Steps** We have conduct the time consumption and tracking performance for different PHC step counts, as presented in Table 18. It can be observed that reducing the number of PHC steps indeed lowers the time cost, but simultaneously leads to a decline in tracking performance. During the retargeting of raw data, we find that the data quality is extremely poor when the number of PHC retargeting steps is insufficient. In addition, we evaluate various metrics of the GMR (Araujo

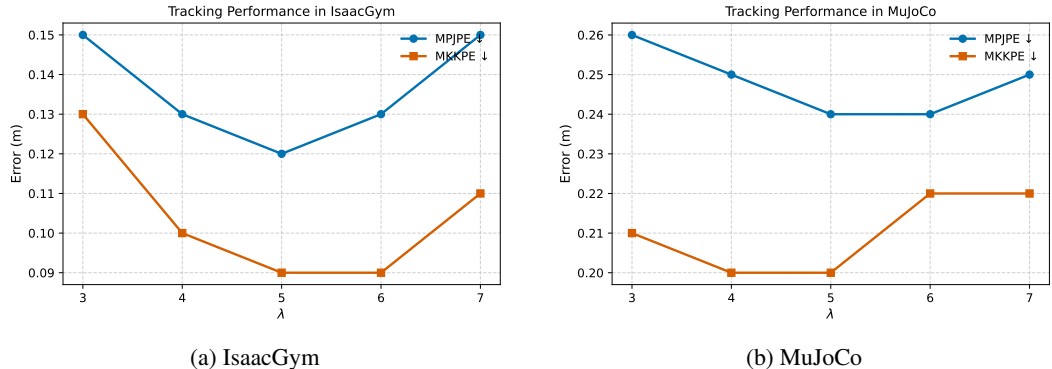

Figure 9: The impact of different number of experts in MoE on tracking performance in IsaacGym and MuJoCo.

| Method | Avg Time (s) $\times 10^{-3}$ | Succ $\uparrow$ | $E_{mpjpe} \downarrow$ | $E_{mpkpe} \downarrow$ |
|---|---|---|---|---|
| DDIM-2 sampling | 4.8 | 0.97 | 0.12 | 0.09 |
| DDIM-4 sampling | 10.7 | 0.97 | 0.12 | 0.09 |
| DDIM-6 sampling | 13.2 | 0.97 | 0.12 | 0.09 |
| DDIM-8 sampling | 17.8 | 0.97 | 0.11 | 0.09 |
| DDIM-10 sampling | 18.1 | 0.97 | 0.11 | 0.08 |

Table 14: Average inference time and tracking performance on different DDIM sampling steps.

et al., 2025) during the testing phase, as shown in Table 18. But we observe frequent joint mutations and distortions for certain martial arts motions when we use GMR to retarget the reference motions.

**Latent from Different Motion Generators**   We conduct ablation studies on two aspects: the impact of latent representations from different frozen motion generators on policy performance, and the effects of finetuning these motion generators on their motion generation metrics as well as policy performance. The settings are as follows: we used three pretrained motion generators: MLD (Chen et al., 2023), MoMask (Guo et al., 2023), and T2M-GPT (Zhang et al., 2023a) as generators of motion latents, and tested two scenarios (finetune and frozen) respectively, evaluating both tracking performance and motion quality metrics on HumanML subsets. As shown in Table 19, we observe that our motion generator achieves the best tracking performance. Furthermore, when we finetune the motion generator, it tends to produce more physically plausible actions, which in turn may lead to a decline in generation metrics, such as MoMask and T2M-GPT. We analyze that the reason may lie in the fact that both MoMask and T2M-GPT are discrete transformer-based motion generators, which need to extract embeddings from the codebook as latents. Specifically, the latents in the codebook are a predefined set of discrete vectors, and during finetuning, the motion generator can only select or combine from the existing discrete candidates. To adapt to the student policy, it may excessively tend to choose latents that are distillation-friendly but incomplete in motion expression; secondly, the number of latents in the codebook is limited, and during finetuning, the motion generator may memorize a small number of latent combinations preferred by the student policy instead of learning

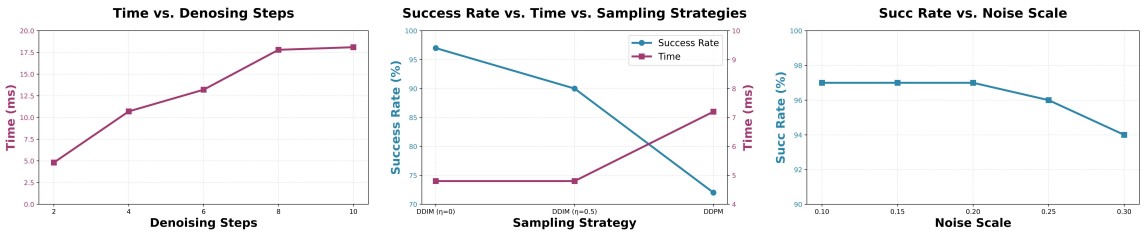

Figure 10: Success rate and latency curves on different settings in diffusion policy.

| Noise Scale ($\beta_{\max}$) | Denoising Steps | Success Rate (%) | Latency (s $\times 10^{-3}$) |
|---|---|---|---|
| 0.10 | 2 | 97.0 | 4.7 |
| 0.15 | 2 | 97.0 | 4.7 |
| 0.20 | 2 | 97.0 | 4.8 |
| 0.25 | 2 | 96.0 | 4.9 |
| 0.30 | 2 | 94.0 | 5.0 |

Note: Fixed settings: 6 denoising steps, DDIM sampling ($\eta$=0), $\beta_{\max}$ denotes the maximum value of $\beta_t$ in 50 training timesteps.

Table 15: Fine-grained ablation on noise scale.

| Sampling Strategy | Denoising Steps | Success Rate (%) | Latency (s $\times 10^{-3}$) |
|---|---|---|---|
| DDIM ($\eta$=0) | 2 | 97.0 | 4.8 |
| DDIM ($\eta$=0.5) | 2 | 90.0 | 4.8 |
| DDPM (Stochastic) | 2 | 72.0 | 7.7 |

Note: Fixed settings: cosine noise schedule (($\beta_{\max}$)=0.20), $\eta$ controls the stochasticity of DDIM sampling.

Table 16: Fine-grained ablation on different sampling strategies

generalized motion laws. This may result in only a few embeddings in the codebook being adjusted, leading to the degradation of generation performance.

## 9.7 PSEUDO CODE

To more clearly present the algorithmic workflow and details of the diffusion student policy, we have provided pseudo code for both the student policy training and the internal operations of the diffusion process, which have been added as Algorithms 1 and 2 in the appendix. Here, we elaborate on the parameters of each component:

The action output frequency of the teacher policy is consistent with that of the student policy. Specifically, during training, the teacher policy needs to be forward-propagated synchronously to generate a supervised action, which is used to supervise the action output by the student policy.

The supervision target of the diffusion process is x0-prediction. Initially, we also attempted to use $\epsilon$-prediction as the optimization target, but its performance was inferior to that of x0-prediction.

The noise scheduling adopts a cosine schedule, with 50 timesteps during training. The noise scale for each timestep is as follows: For the cosine noise schedule with num_diffusion_timesteps = 50, the noise scale ($\beta_t$) for each timestep is computed as follows:

The cumulative product of $(1 - \beta)$ up to normalized timestep $t$ is defined by:

$$\alpha_{\bar{t}} = \cos\left(\frac{t + 0.008}{1.008} \cdot \frac{\pi}{2}\right)^2$$

where $t \in [0, 1]$ is the normalized timestep, calculated as $t = \frac{i}{50}$ for timestep index $i \in \{0, 1, \ldots, 49\}$.

For each timestep $i$ ($0 \leq i \leq 49$), the noise scale is derived as:

$$\beta_i = \min\left(1 - \frac{\alpha_{\bar{t}_2}}{\alpha_{\bar{t}_1}}, 0.999\right)$$

where $t_1 = \frac{i}{50}$ denotes normalized timestep for current step $i$), $t_2 = \frac{i+1}{50}$ denotes normalized timestep for next step $i + 1$, $\alpha_{\bar{t}_1}$ and $\alpha_{\bar{t}_2}$ are the cumulative retention factors at $t_1$ and $t_2$. The upper bound 0.999 avoids numerical singularities.

The conditions consist of proprioceptive information, historical observations, and motion latent representations. Since we use x0-prediction as the optimization target of the diffusion process, the x0-prediction loss can be directly employed as the distillation loss.

| Supervision Target | Succ ↑ | $E_{mpjpe}$ ↓ | $E_{mpkpe}$ ↓ |
|---|---|---|---|
| $\epsilon$-prediction | 0.79 | 0.50 | 0.47 |
| $x0$-prediction | 0.97 | 0.12 | 0.09 |

Table 17: Tracking performance on different optimization objectives.

| Method | Time (s) | Succ ↑ | $E_{mpjpe}$ ↓ | $E_{mpkpe}$ ↓ |
|---|---|---|---|---|
| GMR | 1.41 | 0.92 | 0.23 | 0.18 |
| PHC-100 | 1.63 | 0.81 | 0.45 | 0.40 |
| PHC-500 | 6.09 | 0.88 | 0.31 | 0.25 |
| PHC-800 | 9.87 | 0.91 | 0.25 | 0.21 |
| PHC-1000 | 11.89 | 0.93 | 0.21 | 0.17 |

Table 18: Average inference time and tracking performance on different retargeting methods.

| Method | Top 3↑ | FID↓ | MM-Dist↓ | Succ ↑ | $E_{mpjpe}$ ↓ | $E_{mpkpe}$ ↓ |
|---|---|---|---|---|---|---|
| | | | Frozen | | | |
| MLD | 0.792 | 18.236 | 16.638 | 0.88 | 0.28 | 0.25 |
| T2M-GPT | 0.838 | 12.475 | 16.812 | 0.92 | 0.20 | 0.17 |
| MoMask | 0.846 | 12.232 | 16.138 | 0.95 | 0.16 | 0.14 |
| Ours | 0.867 | 11.706 | 15.978 | 0.97 | 0.12 | 0.09 |
| | | | Finetune | | | |
| MLD | 0.815 | 17.652 | 16.414 | 0.90 | 0.25 | 0.22 |
| T2M-GPT | 0.829 | 12.576 | 16.927 | 0.94 | 0.18 | 0.15 |
| MoMask | 0.849 | 12.376 | 16.233 | 0.96 | 0.14 | 0.12 |
| Ours | 0.878 | 11.529 | 15.884 | 0.98 | 0.09 | 0.07 |

Table 19: Comparison of motion generation metrics and tracking performance under frozen vs. finetuned different motion generators.

---

**Algorithm 1:** DAgger Data Collection and Aggregation for Diffusion Student Policy

---

**Input:** Teacher policy $\pi_{\text{teacher}}$, Student policy $\pi_{\text{student}}$, Environment $\mathcal{E}$, Distillation config $\Omega$,
        Observation dimensions
**Output:** Rollout buffer $\mathcal{B}$
**Initialization**;
Initialize replay buffer $\mathcal{B}$;
obs (including motion latent) $\leftarrow \mathcal{E}$.get_observations();
$\text{obs}_{\text{hist}} \leftarrow \mathcal{E}$.get_extra_hist_obs();
Initialize reward and episode length trackers;
**for** *iter = 1 to $\Omega$.total_iter* **do**
    **Training Phase (Teacher Rollout)**;
    **for** *step = 1 to $\Omega$.num_steps_per_env*(24) **do**
        Aggregate observations with history for buffer $\mathcal{B}$;
        Sample Student Action:
        $\text{action}_{\text{student}} = \pi_{\text{student}}$.sample([obs, obs_hist]);
        Environment step with student action $\text{action}_{\text{student}}$;
        Update episode statistics and historical observations;
    **end**
    **DAgger Data Collection Phase**;
    **for** *step = 0 to $\Omega$.num_mini_batches (4) - 1* **do**
        Prepare observations in the Rollout Buffer: $\text{obs}_{\text{student}} = \mathcal{B}$.pop();
        Prepare observations for teacher:
        $\text{obs}_{\text{teacher}} = \text{obs}_{\text{student}}$[-latent.size(-1):];
        Query teacher for expert actions:
        $\text{action}_{\text{teacher}} = \pi_{\text{teacher}}(\text{obs}_{\text{teacher}})$;
        Student generates actions using diffusion policy ($\mathbf{x}_0$-predicition):
        $\text{action}_{\text{student}} = \pi_{\text{student}}(\text{obs}_{\text{student}})$;
        Compute Distillation Loss:
        $\mathcal{L}_{\text{DAgger}} = \text{MSE}(\text{action}_{\text{student}}, \text{action}_{\text{teacher}})$
    **end**
**end**
**return** $\mathcal{B}$

---

---

**Algorithm 2:** Diffusion Policy with Cosine Schedule and x0-Prediction

---

**Input:** Training dataset $\mathcal{D} = \{(\text{obs}, \text{obs}_{\text{hist}}, z, a^\star)\}$, Precomputed cosine noise schedule $\{\alpha_t\}_{t=0}^T$, Training diffusion steps $T = 50$, Inference DDIM steps $S = 2$, Condition encoder $\mathcal{E}_{\text{cond}}$, Distillation config $\Omega$

**Output:** Trained diffusion policy $\pi_{\text{diff}}$

**Initialization**;

Initialize diffusion model $\mathcal{M}$;

Optimizer $\theta \leftarrow \text{AdamW}(\mathcal{M}.\text{parameters}(), \text{lr}=10^{-4}, \text{weight decay}=10^{-5})$;

**Training Phase**;

**for** *epoch = 1 to $\Omega$.num_epochs* **do**

    **for** *batch in $\mathcal{D}$.shuffle().split($\Omega$.batch_size)* **do**

        $\{(\text{obs}_i, \text{obs}_{\text{hist},i}, z_i, a_i^\star)\} \leftarrow$ batch data;

        `// Step 1: Condition fusion and AdaLN injection`

        $\text{cond}_i \leftarrow \text{concat}(\text{obs}_i, \text{obs}_{\text{hist},i}, z_i) \; \text{cond\_emb}_i \leftarrow \mathcal{E}_{\text{cond}}(\text{cond}_i)$

        `// Step 2: Sample random diffusion timestep and Gaussian`
        `   noise`

        $t \leftarrow \text{UniformSample}(1, T) \; \epsilon \leftarrow \mathcal{N}(0, \mathbf{I})$

        `// Step 3: Forward diffusion (corrupt teacher action with`
        `   noise)`

        $a_t \leftarrow \sqrt{\alpha_t} \cdot a_i^\star + \sqrt{1 - \alpha_t} \cdot \epsilon$

        `// Step 4: Model predicts original action` $a_0$
        `   (x0-prediction)`

        $\hat{a}_0 \leftarrow \mathcal{M}(a_t, t, \text{cond\_emb}_i)$

        `// Step 5: Compute training loss`

        $\mathcal{L} \leftarrow \text{MSE}(\hat{a}_0, a_i^\star)$

        `// Step 6: Backpropagation and parameter update`

        $\nabla_\theta \mathcal{L} \leftarrow \text{ClipGradients}(\nabla_\theta \mathcal{L}, \text{max\_norm} = 1.0); \; \theta \leftarrow \theta - \text{lr} \cdot \nabla_\theta \mathcal{L}$;

    **end**

**end**

**Inference Phase (2-Step DDIM Sampling)**;

`// DDIM sampling pipeline`

$\text{obs}, \text{obs}_{\text{hist}}, z \leftarrow$ Input sensory and latent signals;

`// Step 1: Encode conditions`

$\text{cond} \leftarrow \text{concat}(\text{obs}, \text{obs}_{\text{hist}}, z)$;

$\text{cond\_emb} \leftarrow \mathcal{E}_{\text{cond}}(\text{cond})$;

`// Step 2: Initialize with random noise`

$a_S \leftarrow \mathcal{N}(0, \mathbf{I})$

`// Step 3: 2-step deterministic denoising`

**for** *s = S to 1* **do**

    $t \leftarrow \text{TimestepMap}(s, S, T); \; \hat{a}_0 \leftarrow \mathcal{M}(a_s, t, \text{cond\_emb})$

    `// DDIM update rule (`$\eta = 0$` for deterministic sampling)`

    $\alpha_{t-1} \leftarrow \text{GetPrecomputedAlpha}(t - 1) \; \epsilon_{\text{approx}} \leftarrow \frac{a_s - \sqrt{\alpha_t} \cdot \hat{a}_0}{\sqrt{1 - \alpha_t}}$

    $a_{s-1} \leftarrow \sqrt{\alpha_{t-1}} \cdot \hat{a}_0 + \sqrt{1 - \alpha_{t-1}} \cdot \epsilon_{\text{approx}}$;

**end**

**return** $a_0$

---

