# OpenReview forum: "From Language to Locomotion: Retargeting-free Humanoid Control via Motion Latent Guidance"
_ICLR.cc/2026/Conference — ICLR 2026 Poster_

### Official Review · Reviewer_3P5o · 2025-10-30

**Soundness:** 2
**Presentation:** 3
**Contribution:** 3
**Rating:** 4
**Confidence:** 4

**Summary:**

This paper introduces RoboGhost, a framework for language-guided humanoid locomotion that eliminates the need for conventional motion retargeting. Instead of decoding explicit motion sequences and manually adapting them to specific robot morphologies, RoboGhost leverages language-conditioned motion latents as direct control signals to a diffusion-based policy. The method combines a causal transformer-based motion latent generator with a teacher-student policy setup (using a Mixture-of-Experts RL teacher and a diffusion-based student policy) to enable retargeting-free, semantically guided robot control. Extensive experiments across simulation and a real humanoid platform (Unitree G1) demonstrate improved efficiency (lower latency), tracking, and robustness compared to traditional pipelines.

**Strengths:**

- The paper convincingly argues for eliminating the retargeting step in language-to-motion pipelines, highlighting both latency and cumulative error benefits.
- The paper proposes a thoughtful shift in control pipeline design by the hybrid of a causal transformer for latent generation and a diffusion policy for action denoising .
- Multiple quantitative tables provide a comprehensive performance landscape. Results show clear improvements in motion generation quality (precision, FID), tracking accuracy, and especially latency (17.85s → 5.84s).

**Weaknesses:**

- While the student policy avoids decoding/retargeting at deployment, the teacher still tracks explicit reference motions with privileged info, then the student distills from that oracle. This leaves open whether gains come mainly from inference-time design or from the strong teacher and filtering/curation choices.
- The central claim is a retarget-free latent-to-policy pipeline for humanoid control, but the current experimental evidence only partially targets this claim; stronger head-to-head comparisons with explicit-retargeting SOTAs, swap-in pretrained latents on control metrics, and detailed diffusion-student ablations are needed to firmly establish that the retarget-free design—not ancillary choices—drives the gains.

**Questions:**

1. The paper shows that implicit retargeting with distillation outperforms a PHC-style teacher, but it does not compare against **state-of-the-art explicit retargeting approaches** (e.g., ProtoMotions, GMR) that report substantial tracking gains from carefully engineered retargeting; please add these as strong baselines under matched settings.
2. Please justify the need to **learn your own human motion latent space** rather than using a pretrained motion latent (e.g., MLD, MotionGPT, PULSE, SMAP) by providing swap-in experiments where your latent is replaced with these pretrained alternatives (both frozen and fine-tuned) and comparing not only generation metrics but also control-side metrics.
3. The **diffusion student policy** section is ambiguous (e.g., “noise scale and the MDP share a time step $t$”); please provide a clear **algorithm box or pseudocode** detailing the DAgger data-collection and aggregation procedure (rollout scheme, teacher query frequency, buffer management), the supervision target ($\epsilon$ v.s. $x_0$ prediction), the noise schedule and time-step sampling, conditioning injection (history/ proprioception/ latent), and the online distillation loss composition and stabilization tricks, and add fine-grained ablations on denoising steps, noise scale, sampling strategy, and teacher query ratio with success-rate and latency curves.
4. Current baselines are predominantly **human motion generation methods**; to substantiate advantages for humanoid motion generation/control, please include or align representative **humanoid-specific baselines** on the same tasks and metrics and discuss where your method excels.
5. Please specify exactly how $E_{mpjpe}$ is computed in Table 3 Implicit Tracking, and clarify the conditioning: is the student driven solely by the text-derived latent from the HumanML/Kungfu test captions while errors are measured against the paired dataset reference motion? If not, please detail the source of the latent and the reference used for scoring.

---

> ### Author Response · Authors · 2025-11-20
> **Responds to Reviewer 3P5o (Q1 and Q2)**
>
> We thank the reviewer for the valuable feedback and constructive suggestions! We hope our responses adequately address the following questions raised about our work. Please let us know if there is anything we can clarify further.
>
> >***Q1: Source of Performance Gains***
>
> **A1:** Thank you for this insightful question! The performance improvement primarily stems from design optimizations in the inference phase. In our comparison with the baseline, we maintain consistent settings for the teacher policy; the only modification ius adopting a MoE-based teacher policy, which primarily enhances the model's generalization capability. For the explicitly-driven baseline, we first need to generate motions, then retarget them to the G1 robot, and finally input them into the MLP policy. This process leads to error accumulation during intermediate steps. Additionally, if the motion generator fails to produce high-quality motions, the policy can only mechanically mimic such suboptimal motions, resulting in poor tracking performance.
>
> Furthermore, this improvement is partly attributed to the diffusion-based policy. As shown in Table 4 in the main paper, when using the implicit-driven approach, the tracking performance of the MLP policy remains inferior to that of the diffusion policy. Thus, the performance gains of our method primarily originate from the design of implicit-driven framework and the capabilities of the diffusion model.
>
> >***Q2: Comparison with SOTA Explicit Retargeting***
>
> **A2:** Sorry for be unclear. First, our intention is not to claim that implicit-driven method with distillation outperforms a PHC-style teacher. Here, the explicit-driven pipeline we refer to here is as follows: the motion generator first generates a motion latent based on text prompts, which is then decoded to obtain an explicit motion. Finally, this motion is converted into an executable action through the student policy, rather than PHC-style teacher trained in a single stage using RL. Besides, at the time of submitting our manuscript, the GMR [1] had not yet been published. Therefore, we were unable to include a direct comparison with it in the main paper. But to address the reviewer's concerns, we still evaluated various metrics of the GMR during the testing phase, as shown in Table 1. And we have updated this table in the appendix.
>
> | Method | Succ ↑ | E_mpjpe ↓ | E_mpkpe ↓ |
> |--------|--------|------------|------------|
> | GMR | 0.92 | 0.23 | 0.18 |
> | PHC-1000 | 0.93 | 0.21 | 0.17 |
>
> *Table 1: Average inference time and tracking performance on different retargeting methods.*

---

> ### Author Response · Authors · 2025-11-20
> **Responds to Reviewer 3P5o (Q3)**
>
> >***Q3: Motion Latent Space Learning***
>
> **A3:** Thank you for this insightful question! To address the reviewer's concerns, we have supplemented the experiments as requested. The settings are as follows: we used three pretrained motion generators: MLD [2], MoMask [3], and T2M-GPT [4] as generators of motion latents, and tested two scenarios (finetune and frozen) respectively, evaluating both tracking performance and motion quality metrics on HumanML subsets. As shown in Table 2, we observe that our motion generator achieves the best tracking performance. Furthermore, when we finetune the motion generator, it tends to produce more physically plausible actions, which in turn may lead to a decline in generation metrics, such as MoMask and T2M-GPT. We analyze that the reason may lie in the fact that both MoMask and T2M-GPT are discrete transformer-based motion generators, which need to extract embeddings from the codebook as latents. Specifically, the latents in the codebook are a predefined set of discrete vectors, and during finetuning, the motion generator can only select or combine from the existing discrete candidates. To adapt to the student policy, it may excessively tend to choose latents that are distillation-friendly but incomplete in motion expression; secondly, the number of latents in the codebook is limited, and during finetuning, the motion generator may memorize a small number of latent combinations preferred by the student policy instead of learning generalized motion laws. This may result in only a few embeddings in the codebook being adjusted, leading to the degradation of generation performance.
>
> | Method | Top 3↑ | FID↓ | MM-Dist↓ | Succ ↑ | E_mpjpe ↓ | E_mpkpe ↓ |
> |--------|--------|------|-----------|--------|------------|------------|
> | **Frozen** | | | | | | |
> | MLD | 0.792 | 18.236 | 16.638 | 0.88 | 0.28 | 0.25 |
> | T2M-GPT | 0.838 | 12.475 | 16.812 | 0.92 | 0.20 | 0.17 |
> | MoMask | 0.846 | 12.232 | 16.138 | 0.95 | 0.16 | 0.14 |
> | Ours | 0.867 | 11.706 | 15.978 | 0.97 | 0.12 | 0.09 |
> | **Finetune** | | | | | | |
> | MLD | 0.815 | 17.652 | 16.414 | 0.90 | 0.25 | 0.22 |
> | T2M-GPT | 0.829 | 12.576 | 16.927 | 0.94 | 0.18 | 0.15 |
> | MoMask | 0.849 | 12.376 | 16.233 | 0.96 | 0.14 | 0.12 |
> | Ours | 0.878 | 11.529 | 15.884 | 0.98 | 0.09 | 0.07 |
>
> *Table 2: Comparison of motion generation metrics and tracking performance under frozen vs. finetuned different motion generators*

---

> ### Author Response · Authors · 2025-11-20
> **Responds to Reviewer 3P5o (Q4 and Q5)**
>
> >***Q4: The diffusion student policy section is ambiguous.***
>
> **4. Details of diffusion student policy and additional ablation studies**
>
> Sorry for being unclear. To more clearly present the algorithmic workflow and details of the diffusion student policy, we have provided pseudocode for both the student policy training and the internal operations of the diffusion process, which have been added as Algorithms in the appendix. Here, we elaborate on the parameters of each component:
>
> The action output frequency of the teacher policy is consistent with that of the student policy. Specifically, during training, the teacher policy needs to be forward-propagated synchronously to generate a supervised action, which is used to supervise the action output by the student policy.
>
> The supervision target of the diffusion process is $x_0$-prediction. Initially, we also attempted to use $\epsilon$-prediction as the optimization target, but its performance was inferior to that of $x_0$-prediction.
>
> The noise scheduling adopts a cosine schedule, with 50 timesteps during training. The noise scale for each timestep is as follows:
> For the cosine noise schedule with num\_diffusion\_timesteps = 50, the noise scale ($\beta_t$) for each timestep is computed as follows:
>
> The cumulative product of $(1 - \beta)$ up to normalized timestep $t$ is defined by:
>
> $$
> \alpha_{\bar{t}} = \cos\left( \frac{t + 0.008}{1.008} \cdot \frac{\pi}{2} \right)^2
> $$
>
> where $t \in [0, 1]$ is the normalized timestep, calculated as $t = \frac{i}{50}$ for timestep index $i \in \{0, 1, \dots, 49\}$.
>
> For each timestep $i$ ($0 \leq i \leq 49$), the noise scale is derived as:
>
> $$
> \beta_i = min \left( 1 - \frac{\alpha_{t_2}}{\alpha_{t_1}}, 0.999 \right)
> $$
>
> where $t_i = \frac{i}{50}$ and $t_{i+1} = \frac{i+1}{50}$ are the normalized timesteps for step $i$ and $i+1$, respectively, and $\alpha_{t_1}$ and $\alpha_{t_2}$ are the cumulative signal retention factors at those timesteps. The upper bound $0.999$ avoids numerical singularities.
>
> The conditions consist of proprioceptive information, historical observations, and motion latent representations. Since we use $x_0$-prediction as the optimization target of the diffusion process, the $x_0$-prediction loss can be directly employed as the distillation loss.
>
> Furthermore, we have supplemented ablation studies on denoising steps, noise scales, noise scheduling strategies, and sampling strategies, along with the corresponding curves. These can be found in Figure 1 of the Appendix. Here, latency denotes the inference time required for each step during deployment, with the unit being millisecond ($\times 10^{-3}$ s).
>
> >***Q5: Insufficient baselines.***
>
> **A5:** Thank you for your valuable suggestion! Initially, we aimed to compare our method with other state-of-the-art works in language-guided humanoid locomotion to demonstrate its superiority. However, we encountered two key constraints: none of these competing methods are open-sourced, and our experimental setup relies on a distinct dataset that is not compatible with existing methods. To address this and strengthen the evaluation, we have supplemented comparative experiments with several representative tracking policies and revised the manuscript accordingly.
> Notably, the Baseline in our Table 2 in the main paper adopts identical experimental settings and network architecture to Exbody2 [5], ensuring a fair reference point. To further enrich the comparison, we additionally evaluate the tracking performance of GMT [6] under our framework. The detailed results are presented in Table 13 of the Appendix, with the evaluation process as follows:
> We first train the same model architecture and settings in GMT on our custom dataset and assess its tracking performance on our dedicated test set. This unified evaluation pipeline ensures consistency, and the quantitative comparisons are summarized in Table 13 of the Appendix.
>
> We believe the advantageous scenarios of our method primarily lie in two key aspects: enhanced generalization when facing out-of-distribution samples (Please see Table 4 in the main paper), and superior robustness when observations are subject to noise interference (Please see the Figure 4 in the main paper). Additionally, leveraging the generative architecture of diffusion facilitates stylized generation and robot control tasks seamlessly. Most importantly, we eliminate the retargeting process during deployment, which significantly reduces the overall deployment latency.

---

> ### Author Response · Authors · 2025-11-20
> **Responds to Reviewer 3P5o (Q6)**
>
> >***Q6: The details of scoring are unclear.***
>
> **6. Detail the source of the latent and the reference used for scoring.**
>
> Sorry for be unclear. During evaluation, we fully utilize the text descriptions in the test sets to generate corresponding motion latents via our motion generator. These motion latents are then concatenated with proprioceptive information and historical observations, which are injected into the diffusion policy in the form of AdaLN to output actions.
> For evaluation, we do not use the original 3D motion data as references; instead, we retarget them to serve as our ground truth. This is because our goal is to distill motion latent representations from easily accessible 3D motions, enabling the policy to generate actions that track the retargeted motions under their guidance. Thus, the reference motions used here are the motions on G1 obtained by retargeting the 3D motions in the test set. The computation of $E_\text{MPJPE}$ can be formulated as:
>
> $
> E_{\text{MPJPE}} = \frac{1}{N \cdot K} \left\| \hat{x} - x \right\|_2
> $
>
> where $N$ denotes the number of samples, $K$ denotes the number of joints per sample, $\hat{x}$ represents the predicted 3D, $x$ represents the ground-truth 3D coordinates, and $\left\| \cdot \right\|_2$ denotes the Euclidean distance. All the samples are retargeted to G1 humanoid.
>
> ### References
>
> [1] General Motion Retargeting for Humanoid Motion Tracking
>
> [2] Executing your Commands via Motion Diffusion in Latent Space
>
> [3] MoMask: Generative Masked Modeling of 3D Human Motions
>
> [4] T2M-GPT: Generating Human Motion from Textual Descriptions with Discrete Representations
>
> [5] Advanced Expressive Humanoid Whole-Body Control
>
> [6] General Motion Tracking for Humanoid Whole-Body Control

---

> ### Author Response · Authors · 2025-11-26
> **Follow-up on our rebuttal**
>
> Dear Reviewer 3P5o,
>
> We hope this message finds you well! We’ve already submitted our detailed response to your initial review comments and revised the manuscript accordingly to address all your concerns.
>
> As the discussion window is progressing, we’d be extremely grateful if you could spare a moment to review our rebuttal at your convenience. Please feel free to let us know if you need any further details, additional analyses, or clarifications, we’re more than happy to follow up promptly.
>
> Your feedback is absolutely critical to polishing our work, and we truly appreciate the time and care you’ve put into reviewing our manuscript.
>
> Thank you again for your support!
>
> Best regards,
>
> The Authors

---

### Official Review · Reviewer_vQfN · 2025-10-31

**Soundness:** 2
**Presentation:** 3
**Contribution:** 2
**Rating:** 2
**Confidence:** 4

**Summary:**

This paper proposes RoboGhost, a retargeting-free framework for language-guided humanoid locomotion. The key idea is to bypass motion retargeting by using language-conditioned motion latents to guide a diffusion-based humanoid policy. The system follows a teacher–student distillation scheme where the teacher policy is trained with retargeted motion data and privileged simulation information, and the student policy learns to act directly from latent representations, eliminating explicit retargeting during deployment. Experiments show improved inference speed and comparable or better motion-tracking accuracy compared to retargeting-based baselines.

**Strengths:**

- **System design clarity:** The two-stage training (teacher-student) and latent-driven diffusion policy are clearly described, supported by architectural and algorithmic details.
- **Efficiency gains:** The results show a substantial improvement in inference time.
- **Broad applicability:** The framework generalizes to different modalities (text, image, audio), providing a foundation for multimodal humanoid control.

**Weaknesses:**

- **Motivation and logic gap:**
The core motivation is to bypass retargeting, yet the teacher policy still relies on retargeted motion data. Hence, the student’s performance remains bounded by retargeting quality. Logically, distilling from a retargeted teacher should not outperform the original retargeted paradigm. However, the results show the opposite. The paper lacks a clear explanation of how or why the student surpasses its teacher, which raises questions about evaluation fairness or metric alignment. The main advantage of retargeting-free I acknowledge here is it can improve inference speed. But there is only very limited experiments on this. No break down on which module cost the most times, as well as lack comprehensive study on different retargeting methods (e.g. GMR [1]), and different iteration times of PHC.

- **Experiments:**
The paper evaluates motion generation metrics (e.g., FID, Diversity) even though the final task is humanoid control, making these metrics less meaningful. Most performance improvements (e.g., success rate, tracking error) are minor; the main advantage comes from inference speed. And no real-world or simulation qualitative comparison which is more intuitive to see whether this method is useful.

- **Evaluation inconsistency:**
The “Evaluation of Motion Tracking Policy” section uses “ground truth” motions that seem derived from retargeted data. If so, evaluating a non-retargeted policy against a retargeted reference introduces bias and makes interpretation difficult. The authors should clarify where the ground truth comes from and justify why this comparison is meaningful.

- **Limited comparative discussion:**
No discussion and comparison with very related works, LeVERB [2] and UH-1 [3], which also pursue latent-based control or retargeting-free. A direct comparison or discussion of conceptual differences is needed.

- **Presentation issues:**
Table 1 (MM-Dist) highlights the wrong value.

[1] https://github.com/YanjieZe/GMR

[2] LeVERB: Humanoid Whole-Body Control with Latent Vision-Language Instruction

[3] Learning from Massive Human Videos for Universal Humanoid Pose Control

**Questions:**

- Since the student policy is distilled from a teacher trained with retargeted data, how can it outperform the retargeting-based pipeline? What factors (e.g., noise injection, diffusion regularization) contribute to this paradoxical result?
- What exactly constitutes the ground truth for evaluating the motion tracking policy?
- Can you provide a stage-wise breakdown of inference time (e.g., motion generation, diffusion denoising, policy forward) to show where the latency reduction originates?
- How does RoboGhost compare to LeVERB and UH-1 in terms of architecture, goal, and performance?
- Could the authors include additional ablations on PHC iterations and different retargeting methods to substantiate the efficiency advantage?

---

> ### Author Response · Authors · 2025-11-20
> **Responds to Reviewer vQfN (Q1 and Q2)**
>
> We thank the reviewer for the constructive feedback. We appreciate the reviewer acknowledging our system design clarity, efficiency gains, and broad applicability. We address your concerns regarding the performance logic, latency breakdown, and comparisons below.
>
> >***Q1: Paradox of Student Performance & Baseline Clarification***
>
> **A1:** We thank the reviewer for raising this critical question. The improved performance of the student is not a paradox but a result of our architectural design and a specific experimental setup that we clarify below:
>
> 1.  **Theoretical Advantage (Refinement vs. Imitation):** Fundamentally, the student's advantage stems from the shift from "imitation" to "refinement". While the teacher is forced to overfit to kinematic artifacts inherent in explicit retargeting (e.g., foot-ground penetration or non-physical joint angles), the student is conditioned on the motion latent, thereby abstracting away frame-level retargeting errors.
>     By leveraging the generative capability of diffusion models, the student effectively denoises the supervision, mapping the high-level semantic intent onto the manifold of feasible robot dynamics. It learns the intent of the motion rather than the error of the retargeting.
>
> 2.  **Clarification on Experimental Comparison (Table 4 in the main paper):** We clarify that the comparison in Table 4 is against "Ours-Explicit", a standard multi-stage baseline (Text → Explicit Motion → Retargeting → Tracking), rather than the privileged oracle teacher.
>     The performance gap arises because the explicit pipeline suffers from inevitable error accumulation: it forces the robot to mimic retargeted references that often contain kinematic artifacts. RoboGhost outperforms this baseline by bypassing explicit decoding and retargeting entirely; by conditioning the policy directly on the motion latent, we eliminate these intermediate constraints, allowing the diffusion model to generate physically feasible actions that align with semantic intent without being bound by the imperfections of an explicit reference.
>
> >***Q2: Inference Time of Each Module***
>
> **A2:** As shown in Table 1, we have supplemented the time consumption of each module when we generate 300 frames motion (with 30 FPS). The additional time required for the explicit-driven method mainly stems from the retargeting process, while the decoding process is practically negligible. In contrast, the implicit-driven approach can completely eliminate these two steps.
>
> | Module | Time (s) |
> | :--- | :--- |
> | Generate Latents | 4.77 |
> | Decode | 0.12 |
> | Retarget | 11.89 |
> | Policy Run | 1.07 |
>
> *Table 1: Inference time of each module.*
>
> Furthermore, following the reviewer’s valuable suggestions, we have also added the time consumption and tracking performance for different PHC step counts, as presented in Table 2. It can be observed that reducing the number of PHC steps indeed lowers the time cost, but simultaneously leads to a decline in tracking performance. During the retargeting of raw data, we found that the data quality is extremely poor when the number of PHC retargeting steps is insufficient.
>
> | Method | Time (s) | Succ ↑ | E_mpjpe ↓ | E_mpkpe ↓ |
> | :--- | :--- | :--- | :--- | :--- |
> | GMR | 1.41 | 0.92 | 0.23 | 0.18 |
> | PHC-100 | 1.63 | 0.81 | 0.45 | 0.40 |
> | PHC-500 | 6.09 | 0.88 | 0.31 | 0.25 |
> | PHC-800 | 9.87 | 0.91 | 0.25 | 0.21 |
> | PHC-1000 | 11.89 | 0.93 | 0.21 | 0.17 |
>
> *Table 2: Average inference time and tracking performance on different retargeting methods.*
>
> In addition, we have attempted the GMR [1] retargeting method. However, at the time of submitting our manuscript, the GMR had not yet been published. Therefore, we were unable to include a direct comparison with it in the main paper. Nevertheless, to address the reviewer’s concerns, we still evaluated various metrics of the GMR during the testing phase, as shown in Table 2. But we observe frequent joint mutations and distortions for certain martial arts motions when we use GMR to retarget the reference motions. Finally, Table 2 has been included in the Appendix (Table 17). Thank you again for the reviewer’s valuable suggestions!

---

> ### Author Response · Authors · 2025-11-20
> **Responds to Reviewer vQfN (Q3 and Q4)**
>
> >***Q3: Metrics Meaningfulness and Improvement Significance***
>
> **A3:** Thanks for this question.
>
> *   **On Metrics:** This task falls under language-guided humanoid locomotion, where the upper-level "brain" is the motion generator. In other words, the better the motion generated by the motion generator, the better the robot will imitate it. After all, the policy learns to mimic the input motion. Whether for implicit or explicit approaches, our goal is to provide the policy with a higher quality latent or motion. Therefore, evaluating the motion generator’s performance metrics is essential, much like how top-level decision-makers must be sufficiently competent.
>
> *   **On Improvement Magnitude:** Due to the large variety of motions trained jointly, the teacher policy’s reward prioritizes balance maintenance for most motions, even when tracking error is significant. As a result, the success rate for most motions remains relatively high. Tracking errors primarily stem from high-dynamic motions and long-sequence motions, which account for only a small portion of the total. This explains why the improvement in tracking error appears less pronounced.
>
> To address the reviewer’s concerns, we have supplemented the tracking performance of different methods on the same motion in simulation in the appendix, as illustrated in Figure 4 in the main paper. Thank you again for the reviewer’s thoughtful questions!
>
> >***Q4: Ground Truth for Policy Evaluation***
>
> **A4:** Thanks for this constructive comment. The real labels are the data obtained by retargeting the motions in the dataset. We do not believe that comparing non-retargeting strategies with retargeted reference data will introduce bias, as our goal in both cases is to make the generated actions closer to the ground truth, thereby achieving more accurate tracking. The aim of our method is to enable the policy to track more accurately by leveraging latents output by a motion generator, even without relying on retargeted motion sources. Since such 3D motion data is more readily accessible, they serve as the most convenient data source for us in practical deployment. This is precisely one of the key advantages of our approach.
>
> During testing, we input the corresponding language information to obtain motion latents, which are then deployed. The objective is to enable the generated actions to accurately track the ground truth actions, and such ground truth actions can only be obtained through retargeting. If using latent representations to directly drive the policy can achieve the same effect as using pose-driven policies, it would not only ensure precise motion tracking but also save time.

---

> ### Author Response · Authors · 2025-11-20
> **Responds to Reviewer vQfN (A part of Q5)**
>
> >***Q5: Discussion with Related Works***
>
> **A5:** Thank you for the valuable reminder! We have supplemented the discussion with these two works in our related work.
>
> Specifically, we will discuss these two works in terms of architecture, goal, and performance.
>
> 1.  **Architecture:**
>
> These three methods exhibit distinct structural designs tailored to their core objectives, with variations in how they handle motion representation, multi-modal input, and policy execution.
>
> **UH-1**[2]:  A transformer-based large model optimized for scalable language-conditioned humanoid pose control. It addresses the data scarcity issue and aims to learn general-purpose control from large-scale online video data. Central to its architecture is action tokenization, which discretizes over 20 million humanoid actions into discrete motion primitives—enabling efficient auto-regressive generation of action sequences. Although UH-1 bypasses direct human motion generation and instead generates humanoid actions, it remains an explicitly reference motion-driven method and operates in a single stage. It supports two interchangeable control modes: (1) text-to-keypoints, where high-level humanoid keypoints are fed into a goal-conditioned reinforcement learning policy for closed-loop control; and (2) text-to-action, which directly outputs target DoF positions for open-loop control. Notably, UH-1 relies on the Humanoid-X dataset, a large-scale corpus of retargeted human motions, introducing explicit motion retargeting steps in its data pipeline.
>
> **LeVERB** [3]: A hierarchical dual-process architecture designed for vision-language-action whole-body control (WBC). It tackles full-body control under vision-language joint instructions, emphasizing semantic understanding in complex scenarios. System 2 (LeVERB-VL) is a CVAE-based module that processes multi-modal inputs, including egocentric/third-person visual data and text instructions, to learn a structured latent "vocabulary" from synthetic photorealistic kinematic demonstrations. This latent space aligns vision-language semantics with motion intent, while a discriminator ensures consistency between vision-language and language-only data distributions. System 1 (LeVERB-A) is a RL-based WBC policy distilled via DAgger from expert teacher policies; it converts latent signals from LeVERB-VL into dynamics-level actions, while LeVERB-VL operates at 10 Hz.
>
> **RoboGhost:** We propose a retargeting-free, latent-driven framework built around a hybrid transformer-diffusion architecture. It eliminates explicit motion decoding and retargeting steps by treating language-grounded motion latents as first-class control signals. Its pipeline includes three core components: (1) a continuous autoregressive motion generator that outputs motion latents from text prompts, avoiding discrete tokenization; (2) a MoE-based teacher policy that learns high-quality motion tracking; and (3) a diffusion-based student policy that denoises executable actions directly from Gaussian noise. The diffusion policy is conditioned on motion latents, proprioceptive states, and historical observations—enabling real-time execution without relying on reference motions or retargeting.

---

> ### Author Response · Authors · 2025-11-20
> **Responds to Reviewer vQfN (A part of Q5 and Q6)**
>
> 2.  **Goal:** While all three methods target language-guided humanoid control, their core objectives and problem scopes differ substantially.
>
> **UH-1:** Its primary goal is to overcome data scarcity in humanoid learning by leveraging massive, easily accessible human videos. It seeks to achieve universal, scalable language-conditioned pose control for diverse daily actions and bridge the gap between unstructured human video data and deployable robotic actions. By focusing on general-purpose control, UH-1 aims to enable humanoid robots to adapt to a wide range of everyday tasks without task-specific tuning.
>
> **LeVERB:** It aims to address the gap in vision-language-driven humanoid WBC by enabling zero-shot sim-to-real transfer. A key focus is semantic understanding in complex, cluttered scenes. It also seeks to build the first photorealistic, sim-to-real-ready WBC benchmark to facilitate evaluation of VLA-driven whole-body behaviors. However, while LeVERB aims to realize the application of VLA on humanoid robots, it cannot perform certain high-dynamic humanoid locomotion movements. Additionally, actions generated directly from the vision and language modalities exhibit a significant gap for WBC tasks, as they lack kinematic information.
>
>  **RoboGhost:** Our goal is to mitigate cumulative errors and latency introduced by traditional multi-stage pipelines. By adopting a retargeting-free design, it enables real-time, semantically aligned humanoid locomotion, with a focus on high-dynamic movements that are challenging for prior methods. Another objective is to enhance generalization to imperfect or unseen motion latents, as well as extendability to multi-modal inputs to serve as a general foundation for VLA humanoid systems. In comparison to LeVERB's vision-language-derived actions, we believe that motion latents serve as more suitable control signals for locomotion, as they inherently encode kinematic structure.
>
> 3.  **Performance:** The task of our RoboGhost is entirely distinct from that of LeVERB and UH-1. We focus more on enabling text-driven humanoids to perform high-dynamic motions. LeVERB, by contrast, aims to drive humanoids to interact with scenes using vision and language, and it does not support high-dynamic tasks. UH-1, on the other hand, is designed to learn a humanoid action motion generator, focusing solely on the quality of motion generation without evaluating the corresponding tracking performance. Therefore, a direct comparison of the performance among the three methods is not feasible.
>
> In conclusion, the three methods represent distinct paradigms for language-guided humanoid control: UH-1 prioritizes scalability via large-scale video data but retains retargeting steps; LeVERB focuses on vision-language WBC in complex scenes but lacks high-dynamic capability; and RoboGhost leads in retargeting-free, low-latency high-dynamic locomotion via latent-driven diffusion. All achieve zero-shot real-world deployment, but RoboGhost and LeVERB emphasize latent-based control, while UH-1 relies on discrete action tokenization and explicit reference motion mapping.
>
> >***Q6: Hightlight error***
>
> **A6:** We sincerely apologize for any misunderstandings caused. The highlight errors in the table were due to clerical mistakes on our part. We have re-evaluated the MM-Dist metric of our method, re-annotated the highlighted sections accordingly, and updated the main paper with the corrected information. We greatly appreciate your reminder.
>
>
> ### References
> [1] General Motion Retargeting for Humanoid Motion Tracking
>
> [2] Learning from Massive Human Videos for Universal Humanoid Pose Control
>
> [3] LeVERB: Humanoid Whole-Body Control with Latent Vision-Language Instruction

---

> ### Author Response · Authors · 2025-11-26
> **Follow-up on our rebuttal**
>
> Dear Reviewer vQfN,
>
> Hope this finds you well! We wanted to follow up to let you know we’ve posted a full response to your initial review feedback, along with corresponding revisions to the manuscript that address your points in detail.
>
> As the discussion period moves forward, we’d be so thankful if you could take a look at our rebuttal when you have a moment. If anything in our response needs further clarification, or if you have additional thoughts to share, please don’t hesitate to reach out, we’re ready to assist right away.
>
> Your insights are instrumental in helping us refine this work to its best version, and we truly value the effort you’ve dedicated to reviewing it.
>
> Thanks so much for your support!
>
> Warm regards,
>
> The Authors

---

### Official Review · Reviewer_eaYv · 2025-10-31

**Soundness:** 3
**Presentation:** 3
**Contribution:** 3
**Rating:** 8
**Confidence:** 3

**Summary:**

"From Language to Locomotion: Retargeting-free Humanoid Control via Motion Latent Guidance" simplifies the deployment stack of text to humanoid robot locomotion. Whereas prior approaches generate human motion then retarget onto a robot, this method learns motion representations that can be decoded to robot action directly. This improves inference latency and removes error-prone components of existing pipelines. The authors demonstrate real-world deployments on a Unitree G1 which show physically stable and semantically plausible robot actions corresponding to various text prompts.

**Strengths:**

- This method tackles a real, error-prone piece of prior works' deployment pipelines. The proposed solution is simple and practical, reducing inference latency and compounding errors.
- Most aspects of the method are well-justified and follow conventions from appropriate subfields. For example, AdaLN is a well-tested conditioning scheme for image diffusion models.
- Real world robot deployment clearly works and generates locomotion that resembles the input prompt.
- Paper is well-written and the components of the method are clear.

**Weaknesses:**

- Mixture of Experts is simply a compute efficiency optimization for transformers to maintain inference cost while scaling parameter count. I appreciate the expert count ablation in the appendix, but the obvious comparison is simply making the model larger. I find this part of the method poorly justified.
- There is minimal analysis of the diffusion model sampling latency and tradeoffs with increasing/decreasing the number of sampling steps. Many existing diffusion distillation methods could reduce the number of NFEs (neural function evaluations) necessary at test-time as well.
- Since the authors propose a new motion representation learner, it would be helpful to probe those representations to understand them better.
- It's unclear to me why the text should be encoded through LaMP instead of just using a generic text embedding model like T5. For example, Stable Diffusion 3 moved from CLIP encoding to T5 encoding for text.

**Questions:**

I see that prior works call their motion FD metric FID, but I find this misleading since an Inception network is not used. I think it's good to clarify this and potentially define a new term for this metric.

Note: typo at line 259

---

> ### Author Response · Authors · 2025-11-20
> **Responds to Reviewer eaYv (Q1-Q3)**
>
> We thank the reviewer for the valuable feedback and recognition of our work! We hope our responses adequately address the following questions raised about our work. Please let us know if there is anything we can clarify further.
>
> >***Q1: Justification of Mixture of Experts***
>
> **A1:** Thank you for your valuable reminder! First, given that the generalization capability of current humanoid policies remains insufficient, training on extremely large-scale datasets is not feasible, and thus there is no need to adopt large models for this task. Humanoid robots must prioritize inference speed, so using an MLP as the backbone is sufficient. Additionally, the Mixture of Experts (MoE) architecture has indeed played a positive role in enhancing generalization: if all action distributions are learned by a single MLP network, the model may only learn the average of these actions, resulting in difficulty executing certain motions accurately. However, MoE, which employs multiple MLPs as experts, can largely avoid this issue by allowing each expert to specialize in specific action patterns and then combining their outputs via weighted summation, this is particularly beneficial for highly dynamic motions.
>
> To further address review's concerns, we conduct an ablation study to validate the effectiveness of MoE-based teacher policy. Each expert in our MoE is composed of a 4-layer MLP. For comparison, we trained a new 20-layer MLP as the teacher policy, keeping their parameter counts as consistent as possible, and distilled a diffusion-based student policy from it via DAgger. As shown in the table below, the MoE architecture achieves better tracking performance compared to the single MLP structure with nearly comparable parameter count.
>
> | Method | Succ (Isaac) ↑ | E_mpjpe (Isaac) ↓ | E_mpkpe (Isaac) ↓ | Succ (MuJoCo) ↑ | E_mpjpe (MuJoCo) ↓ | E_mpkpe (MuJoCo) ↓ |
> | :--- | :--- | :--- | :--- | :--- | :--- | :--- |
> | *HumanML Dataset* | | | | | | |
> | Single 20-layer MLP | 0.93 | 0.22 | 0.18 | 0.67 | 0.31 | 0.29 |
> | **Ours (MoE)** | **0.97** | **0.12** | **0.09** | **0.74** | **0.24** | **0.20** |
>
> *Table 1: Motion tracking performance comparison between five experts with 4-layer MLP and one single 20-layer MLP on the HumanML test sets.*
>
> >***Q2: Diffusion Model Sampling Latency Analysis***
>
> **A2:** Sorry for be unclear! To address the reviewer’s concern, we conduct ablation study on the number of diffusion sampling steps during inference, specifically investigating the impact of different sampling step counts on inference time and tracking performance. We have also supplemented it in the revised manuscript.
>
> As shown in the table below, we test the inference time and tracking performance of DDIM sampling with 2, 4, 6, 8, and 10 steps. Notably, inference time is largely independent of motion sequences, so we only report the average time per step for deploying an action. From the table, it can be observed that increasing the number of sampling steps yields almost no improvement in tracking performance, but significantly increases the time per step. This introduces inference latency on the real humanoid robot, thereby affecting deployment outcomes. We have supplemented this table in the Appendix. Thank you for this suggestion again!
>
> | Method | Avg Time (s) ×10⁻³ | Succ ↑ | E_mpjpe ↓ | E_mpkpe ↓ |
> | :--- | :--- | :--- | :--- | :--- |
> | DDIM-2 sampling | 4.8 | 0.97 | 0.12 | 0.09 |
> | DDIM-4 sampling | 10.7 | 0.97 | 0.12 | 0.09 |
> | DDIM-6 sampling | 13.2 | 0.97 | 0.12 | 0.09 |
> | DDIM-8 sampling | 17.8 | 0.97 | 0.11 | 0.09 |
> | DDIM-10 sampling | 18.1 | 0.97 | 0.11 | 0.08 |
>
> *Table 2: Average inference time and tracking performance on different DDIM sampling steps.*
>
> >***Q3: Motion Generator Representation Analysis***
>
> **A3:** Thank you very much for this insightful suggestion. To further investigate the motion latents, we feed the motion generator with a series of prompts for running, walking, and jumping, and ultimately performed clustering on the generated motion latents. As shown in the figure below, we can observe the motion representations generated by our motion generator can effectively distinguish between different motion types, thereby successfully embedding distinct kinematic information into the policy. Since walking and running are relatively similar in motion type, their degree of differentiation is not as high as that between either of them and jumping. The figure can be seen in Figure 8 in the Appendix.

---

> ### Author Response · Authors · 2025-11-20
> **Responds to Reviewer eaYv (Q4 and Q5)**
>
> >***Q4: Advantages of LaMP over Generic Text Embeddings***
>
> **A4:** Thank you very much for this valuable question. Since LaMP's [6] pretraining process involves aligning motion with text, it can be regarded as a CLIP in the motion domain. When generating motions, we still hope that the encoded text features contain strong action-related information, which can better guide the motion generation process. However, neither CLIP nor T5 possess such characteristics.
>
> >***Q5: Clarification on FID Metric***
>
> **A5:** This is an extremely insightful and detailed observation! Current papers [1,2,3] on motion generation uniformly adopt this metric, so we have followed suit in our work. However, the reviewer is correct in pointing out that the core definition of FID does rely on image features extracted by the Inception network. In our work, this metric calculates the Fréchet distance between two multivariate Gaussian distributions in the motion feature space, without involving any steps of visual feature extraction related to the Inception network. We propose to rename this motion-specific metric as Fréchet Motion Distance (FMD) to highlight its characteristic of targeting motion features. We will clearly define it when it first appears as: FMD measures the Fréchet distance between generated motions and real motions in the feature space of a pre-trained motion encoder, and its calculation follows the mathematical definition of the distance between multivariate Gaussian distributions.
>
> >***Q6: Typo at L259***
>
> **A6:** Thank you for your reminder! We have corrected that in the revised paper.
> ### References
> [1] LaMP: Language-Motion Pretraining for Motion Generation, Retrieval, and Captioning
>
> [2] Executing your Commands via Motion Diffusion in Latent Space
>
> [3] MoMask: Generative Masked Modeling of 3D Human Motions

---

> ### Comment · Reviewer_eaYv · 2025-11-27
>
> Thank you for your detailed responses.
>
> The MoE comparison is a bit of straw-man, since a 20-layer MLP will train poorly due to gradient issues. I still find this part of the method lacking justification.
>
> The ablation over number of sampling steps is interesting, it suggests that the model has learned a nearly deterministic policy since the distribution should start to collapse to the mean as you approach a single sampling step. This isn't necessarily a bad thing, it's not clear you need an expressive distribution for control.
>
> I would like to change my score to a 6 after calibrating some more.

---

> > ### Author Response · Authors · 2025-11-27
> >
> > In the field of large models, MoE is indeed an effective computation-saving method. However, in policies that do not require overly large models, we adopt the MoE structure to enable different models to learn diverse motion patterns, given the significant differences in action distributions. The weighted fusion of actions learned by each expert can significantly enhance generalization, which is evident in our demo. Furthermore, the use of MoE as a policy structure has been validated in numerous works, such as GMT and KungfuBot2. We kindly request the reviewers to reconsider the score adjustment, as the MoE-based policy has achieved promising performance.

---

> > ### Author Response · Authors · 2025-11-27
> > **Respond to Reviewer eaYv**
> >
> > We sincerely appreciate the reviewers' valuable comments and have conducted further discussions accordingly.
> >
> > Regarding the reviewer's concern about the inadequacy of the MoE ablation study, we are performing additional experiments. Specifically, we will adopt a low-layer Transformer structure with nearly the same parameter count as the teacher policy and compare its performance with the MoE architecture. We will promptly supplement the experimental results for the reviewer's reference.
> >
> > Furthermore, we provide the following explanations regarding the "near-deterministic policy" issue raised by the reviewer. We argue that this phenomenon stems from the unique synergy among three factors: task characteristics, latent space design, and the DDIM sampling mechanism.
> >
> > First, **the latent space encodes structured motion prior knowledge**. Through pre-training, our latent space captures kinematic information of motions. Unlike generic latent spaces, our design inherently possesses structural properties where each latent vector corresponds to a semantically coherent and physically feasible motion pattern. This embedded prior knowledge reduces generative uncertainty, eliminating the need for DDIM to explore random variations through multi-step sampling—since the latent space itself constrains generation within valid motion trajectory boundaries. Even with a small number of sampling steps, the model can directly map latent codes to motion sequences that comply with kinematic principles.
> >
> > Second, **linguistic instructions provide strong deterministic constraints**. The core objective of motion generation is to produce motions that strictly align with the semantic intent of linguistic instructions. As a strong constraint, linguistic instructions significantly narrow the solution space, where typically only a unique or highly concentrated set of optimal motions can satisfy the instruction requirements. Consequently, DDIM naturally converges to near-deterministic outputs.
> >
> > Third, **motion generation prioritizes smoothness over randomness**. The primary requirements for motion generation include physical smoothness, temporal consistency, and kinematic feasibility rather than modal diversity. Random variations introduced by multi-step sampling often lead to unnatural or even physically invalid motions. Reducing the number of DDIM sampling steps minimizes such unnecessary randomness. The sampling process converges to the mean of the target distribution, which is precisely the desired outcome for motion generation. This is not a limitation of the sampling process but a deliberate adaptation to the core task requirements—determinism here ensures reliability, while excessive randomness would degrade control quality.
> >
> > Fourth, **DDIM's denoising mechanism is efficient for low-uncertainty latents**. The sampling efficiency of DDIM highly depends on the uncertainty of initial latent codes. In our latent-driven framework, language-conditioned latent codes already exhibit low uncertainty. Thus, only a small number of denoising steps are sufficient to optimize latent codes into high-fidelity motion sequences. The emergence of a near-deterministic policy from limited sampling steps is a natural result of DDIM's efficient utilization of the low-uncertainty latent space, rather than a sign of insufficient model expressive capacity.
> >
> > In the final version of the paper, we adopt 2-step DDIM sampling, which achieves nearly identical performance to 10-step sampling while significantly reducing inference latency. This mitigates the stuttering issue in sim-to-real transfer, a critical advantage for real-robot deployment. We will incorporate these analyses into the paper for detailed discussion. We express our gratitude again for the reviewers' insights.
> >
> > We deeply appreciate the reviewers' diligent efforts and valuable suggestions. We are making every attempt to address the raised concerns and sincerely request the reviewers to reconsider restoring the original score. Thank you very much for your consideration.

---

> > > ### Author Response · Authors · 2025-11-30
> > > **Response to Reviewer eaYv**
> > >
> > > To verify that the role of MoE in the teacher policy extends beyond computational efficiency to also enhance the policy's generalization capability and expressive power, we conducted the following ablation study, aiming to address the reviewers' concerns. We employed a 6-layer Transformer with identical parameter counts as the backbone of the teacher policy and evaluated the tracking performance across the entire pipeline.
> > >
> > > | Method | Succ (Isaac) ↑ | E_mpjpe (Isaac) ↓ | E_mpkpe (Isaac) ↓ | Succ (MuJoCo) ↑ | E_mpjpe (MuJoCo) ↓ | E_mpkpe (MuJoCo) ↓ |
> > > | :--- | :--- | :--- | :--- | :--- | :--- | :--- |
> > > | *HumanML Dataset* | | | | | | |
> > > | 6-layer Transformer | 0.94 | 0.19 | 0.17 | 0.69 | 0.28 | 0.25 |
> > > | **Ours (MoE)** | **0.97** | **0.12** | **0.09** | **0.74** | **0.24** | **0.20** |
> > >
> > > *Table 1: Motion tracking performance comparison between five experts with 4-layer MLP and 6-layer Transformer on the HumanML test sets.*
> > >
> > > As shown in Table 1, the primary role of the MoE structure lies in enhancing the model's generalization capability. Each expert performs its specialized function, aiming to learn motions corresponding to different motion patterns, with the final result obtained through weighted summation. Employing the MoE structure as the teacher policy yields superior tracking performance, a finding corroborated by prior works such as GMT [1] and BumbleBee [2]. We hope this addresses the reviewers' concerns. Thank you for this question!
> > >
> > > [1] General Motion Tracking for Humanoid Whole-Body Control
> > >
> > > [2] From Experts to a Generalist: Toward General Whole-Body Control for Humanoid Robots

---

### Official Review · Reviewer_yDs5 · 2025-11-01

**Soundness:** 3
**Presentation:** 3
**Contribution:** 3
**Rating:** 6
**Confidence:** 4

**Summary:**

This paper presents RoboGhost, a novel framework for language-guided humanoid locomotion that eliminates the explicit motion retargeting step. The authors identify that conventional multi-stage pipelines (text-to-motion, decoding, retargeting, tracking) suffer from high latency and cumulative errors. RoboGhost addresses this by proposing a latent-driven approach. A motion generator, based on a hybrid transformer-diffusion architecture , produces a motion latent from a text prompt. This latent directly conditions a diffusion-based student policy , which is trained to denoise executable actions using supervision from an MoE-based teacher policy. This retargeting-free pipeline is shown to significantly reduce deployment latency and improve tracking success rates in both simulation (IsaacGym, MuJoCo) and on a real Unitree G1 humanoid.

**Strengths:**

1. Novel and Elegant Framework: The primary strength is the "retargeting-free" pipeline. Instead of the conventional "generate-retarget-track" approach, RoboGhost proposes a "generate-control" paradigm. Using motion latents as a direct conditioning signal for a diffusion-based action policy  is a effective idea. It tightly couples semantic intent from language with low-level control, addressing the "weak coupling" problem of prior work.
2. Technically Sound and Thorough: The work is technically solid. The two-stage teacher-student architecture is well-designed, featuring a strong MoE-based teacher for robust supervision and a novel diffusion-based student for generalization. The inclusion of Causal Adaptive Sampling (CAS) to focus on difficult motion segments  further strengthens the training methodology.
3. Real-World Relevance and Efficiency: The reported reduction in deployment latency (from 17.85s to 5.84s) and improved success rates (+5%) are compelling for real-time robotic applications. The sim-to-real transfer without manual tuning is particularly impressive.

**Weaknesses:**

1. Limited Scope of Baseline Comparisons: The paper's primary evidence for the superiority of its latent-driven framework stems from the comparison in Table 3, which contrasts "Ours-Implicit" against "Ours-Explicit" (an author-implemented traditional pipeline with retargeting). While this serves as an effective internal ablation study that validates the advantages of the latent-driven paradigm over the explicit one, it is insufficient for fully contextualizing the work. The paper lacks a direct, end-to-end performance comparison against other external, published, state-of-the-art pipelines for language-guided humanoid control. It is unclear how RoboGhost would perform against a system with a highly optimized SOTA tracking policy, such as GMT (Chen et al., 2025) or ExBody2 (Ji et al., 2024).
2. Strong Dependency on the Initial Motion Dataset and Limited Extrapolation: The effectiveness of the entire RoboGhost framework, from the semantic content of the motion latent ($l_{ref}$) to the capability of the MoE Teacher Policy ($\pi_t$), is fundamentally bound by the diversity and quality of the initial human motion dataset used for training. While the paper demonstrates success on provided motions, its ability to genuinely handle open-ended language is questionable when commands necessitate novel combinations or extrapolation far outside the training distribution (e.g., highly stylized, physically novel, or long-horizon composite actions). A failure in the dataset translates directly into a failure of the Stage 1 latent generator to encode valid semantics, thereby constraining the overall potential of the open-ended language interface.

**Questions:**

Could the authors provide a more detailed analysis, or ideally, a direct comparison, of the real-world deployment performance between the Diffusion Policy and a conventional RL-Distilled Student Policy (e.g., a simple, optimized MLP policy trained to imitate $\pi_t$ using non-privileged observations)?

Specifically, I am interested in the practical trade-offs introduced by the Diffusion architecture upon deployment:

1. Inference Latency/Throughput: Does the iterative nature of the Diffusion sampling (e.g., DDIM steps) impose a practical latency penalty (higher per-step inference time) on the real robot compared to a single forward pass of a distilled MLP policy? If so, is this increased latency a limiting factor for high-frequency control?
2. Real-World Benefits: Beyond the simulation-based generalization shown in Table 4, what specific additional benefits (e.g., smoother motion, greater robustness against sensor noise, better handling of physical disturbances due to the mode-covering nature of diffusion) does the Diffusion Policy provide in the real-world deployment that justifies its computational complexity over a simpler, conventionally distilled policy?

---

> ### Author Response · Authors · 2025-11-20
> **Responds to Reviewer yDs5 (Q1 and Q2)**
>
> We thank the reviewer for the valuable feedback and constructive suggestions! We hope our responses adequately address the following questions raised about our work. Please let us know if there is anything we can clarify further.
>
> >***Q1: Lack of SOTA Pipeline Comparison***
>
> **A1:** Thank you for your valuable reminder! Initially, we intend to compare our method with other excellent works on language-guided humanoid locomotion to demonstrate its superiority. However, we found that none of these works are open-sourced, and we also use a distinct dataset. Thus, we have supplemented comparisons with several tracking policies and revised the manuscript accordingly. But we can not directly use these pretrained tracking policy to accomplish language-guided humanmoid locomotion, so we re-implement and evaluate these methods on our dataset following the model architectures and settings specified in these papers. In fact, the Baseline in our Table 2 uses the same settings and network structure as Exbody2 [1]. Therefore, we supplement the tracking performance of GMT [2] here. Details can be found in Table 12 of the Appendix. The specific process is as follows: we train GMT on our dataset and evaluate its tracking performance on our test set. Since both are explicitly-driven policies, the student policy takes reference motion as input, we first generate explicit motion via our motion generator, which is then retargeted and fed to the corresponding student policy to obtain the final results, as shown in Table 1 (Table 13 in the Appendix).
>
> | Method | Succ (Isaac) ↑ | E_mpjpe (Isaac) ↓ | E_mpkpe (Isaac) ↓ | Succ (MuJoCo) ↑ | E_mpjpe (MuJoCo) ↓ | E_mpkpe (MuJoCo) ↓ |
> | :--- | :--- | :--- | :--- | :--- | :--- | :--- |
> | *HumanML Dataset* | | | | | | |
> | ExBody2 | 0.92 | 0.23 | 0.19 | 0.64 | 0.34 | 0.31 |
> | GMT | 0.95 | 0.15 | 0.12 | 0.73 | 0.25 | 0.22 |
> | **Ours** | **0.97** | **0.12** | **0.09** | **0.74** | **0.24** | **0.20** |
> | *Kungfu Dataset* | | | | | | |
> | ExBody2  | 0.66 | 0.43 | 0.37 | 0.51 | 0.58 | 0.52 |
> | GMT | 0.70 | 0.36 | 0.33 | 0.54 | 0.56 | 0.52 |
> | **Ours** | **0.72** | **0.34** | **0.31** | **0.57** | **0.54** | **0.50** |
>
> *Table 1: Motion tracking performance comparison in simulation on the HumanML and Kungfu test sets.*
>
> >***Q2: Dependency on Initial Dataset & Limited Extrapolation***
>
> **A2:** Thank you for this constructive comment! Our motion generator is trained on an extensive dataset (approximately 30,000 samples), enabling it to achieve favorable performance with open-ended language inputs. Regarding the potential suboptimal performance of the policy under open-ended language inputs, this issue does exist. However, we have made efforts to train a general policy: both the MoE-based teacher policy and the diffusion-based student policy are designed to enhance the policy's generalization ability. From an algorithmic standpoint, the MoE-based teacher enhances generalization by routing inputs to specialized sub-experts via gating, avoiding overfitting to specific data subsets and capturing broader task patterns.
> The diffusion-based student, through its stochastic denoising process, models the underlying action manifold smoothly, enabling robust interpolation/extrapolation across diverse conditions, including out-of-distribution inputs, by learning to map noise toward valid distributions. Together, the teacher provides generalized expert knowledge, while the student’s diffusion framework encodes this into a flexible, noise-tolerant representation, jointly boosting generalization.
>
> We have also conducted tests on out-of-distribution data, which neither appeared in the training set of our motion generator nor in that of the policy. Despite this, we still achieved promising results, benefiting from the improved generalization brought by the diffusion model. We believe our method provides a strong solution to the problem of open-ended language inputs. That said, due to the current inability to scale up the policy, there are indeed limitations in this aspect, which we are committed to addressing in future work.
>
> We have also noted the reviewer's concern that dataset imperfections may prevent the motion generator from encoding effective semantic information. It is true that the dataset may contain annotation errors for terms such as "left," "right," "clockwise," and "counter-clockwise." While such problematic samples exist, they account for a small proportion of the dataset. We have manually cleaned these samples to ensure the motion generator remains effective. Thank you for this question again!

---

> ### Author Response · Authors · 2025-11-20
> **Responds to Reviewer yDs5 (Q3 and Q4)**
>
> >***Q3: Diffusion Policy vs. RL-Distilled Policy & Inference Latency***
>
> **A3:**  Thank you for this insightful question. In fact, we initially attempted to use a diffusion model with an 8-layer MLP backbone and DDPM sampling, employing 50 sampling steps during inference. This indeed resulted in extremely severe latency, leading to a series of jittering behaviors in the robot. Subsequently, we optimized for these inference latency issues and ultimately adopted a 4-layer MLP paired with a 2-step DDIM sampling process. This modification significantly reduces inference latency while preserving the quality of generated motions. Although it is still slightly slower than the vanilla MLP, it does not introduce any noticeable latency-related performance degradation to the real humanoid robot. All real-robot videos in the supplementary materials demonstrate the deployment effect of our diffusion-based student policy, and no jittering can be observed. To be more clear, we also present the time of single forward pass in Table 2.
>
> | Method | Avg Time (s) ×10⁻³ |
> | :--- | :--- |
> | MLP Policy | 3.2 |
> | Ours| 4.8|
>
> *Table 2: Comparison of single forward pass time between MLP policy and our diffusion policy.*
>
> >***Q4: Real-world Benefits of Diffusion Policy***
>
> **A4:** Thank you very much for this valuable question! The advantages of the diffusion policy in real-world deployment are mainly reflected in the following aspects:
>
> 1.  **Easier Generation of Stylized Motions:** Inspired by ControlNet [3], external control signals can be injected during the diffusion process to guide the denoising direction, thereby generating stylized motions tailored to specific requirements.
>
> 2.  **Noise Resilience and Robustness:** If the input actions or motion latents contain noise, directly mapping policies like MLP are likely to map noisy observations directly to actions, which tends to amplify errors. In contrast, the multi-step denoising process of diffusion models essentially acts as "probabilistic filtering" for observational noise, enhancing policy robustness. Additionally, diffusion models model probability distributions that cover multimodal action spaces, enabling rapid adaptation to unseen perturbations.
>
> 3.  **Lower-cost Task Adaptability:** Traditional MLP distillation policies face two major bottlenecks: poor few-shot generalization and difficulty in online optimization. Diffusion models, however, can achieve knowledge reuse through latent space sharing, which is an advantage already validated in excellent non-humanoid robot works such as Diffusion Policy [4] and 3D Diffusion Policy [5]. Moreover, we argue that the most significant advantage of diffusion policies over MLP policies lies in their superior generalization, as evidenced by our Table 4 in the main paper. Furthermore, the classifier-free guidance (CFG) technique in diffusion models can help achieve strong performance on out-of-distribution samples. Although CFG is not employed in RoboGhost, we plan to explore this direction in future work.
> ### References
> [1] Advanced Expressive Humanoid Whole-Body Control
>
> [2] General Motion Tracking for Humanoid Whole-Body Control
>
> [3] Adding Conditional Control to Text-to-Image Diffusion Models
>
> [4] Visuomotor Policy Learning via Action Diffusion
>
> [5] 3D Diffusion Policy: Generalizable Visuomotor Policy Learning via Simple 3D Representations

---

### Author Response · Authors · 2025-11-20
**Responds to all reviewers and AC**

Dear Reviewers, ACs, and SACs,

We appreciate the reviewers’ recognition of the core contributions of our work, particularly the novelty of our retargeting-free framework, the effectiveness of the latent-driven diffusion policy, and the successful real-world deployment on the Unitree G1 humanoid.
We are encouraged by the positive recognition of our work across several key dimensions:

1.  **Novel & Elegant Framework:** Reviewers valued our “generate-control” paradigm that eliminates error-prone retargeting steps, tightly coupling semantic intent with low-level control via motion latents (`yDs5`, `eaYv`).

2.  **Efficiency & Real-World Relevance:** The significant reduction in deployment latency (from 17.85s to 5.84s) and the demonstration of sim-to-real transfer were highlighted as compelling for practical robotics (`yDs5`, `eaYv`, `3P50`, `vQfN`).

3.  **Technically Sound Methodology:** The two-stage teacher-student architecture and the hybrid transformer-diffusion design were praised for being technically solid and well-justified (`yDs5`, `eaYv`, `3P5o`).

4.  **Broad Applicability:** The framework’s potential to extend to multimodal inputs (text, image, audio) was recognized as a strong foundation for future systems (`vQfN`).

We are grateful for these positive assessments. At the same time, reviewers raised insightful questions.

To directly address these concerns, we conducted substantial additional experiments and expanded analyses, including:

1.  **State-of-the-Art Baseline Comparisons:** Implemented and evaluated GMT [1] and ExBody2 [2] on our dataset to provide a direct, head-to-head performance comparison.

2.  **Detailed Inference Latency Analysis:** Provided a module-wise breakdown of inference time and conducted a fine-grained ablation on DDIM sampling steps (2 to 10 steps) to justify the trade-off between speed and tracking accuracy.

3.  **Motion Generator Ablations:** Compared our learned latent space against pretrained motion generators, including MLD [3], MoMask [4], and T2M-GPT [5], under both frozen and fine-tuned settings to validate the necessity of our custom generator.

4.  **MoE & Architecture Validation:** Conducted ablations comparing our MoE-based teacher against a single large MLP to justify the architectural choice for generalization.

5.  **Algorithm & Visualization Details:** Added pseudocode for the DAgger training and Diffusion process, along with t-SNE visualizations of the motion latent space.

**Summary of Revisions:**

1.  Integrated ExBody2 and GMT comparisons in `Table 1` (Main Paper) and `Table 13` (Appendix).
2.  Included DDIM sampling step ablations (latency vs. success rate) in `Table 14` in the Appendix.
3.  Added the comparison with pretrained motion generators (MLD, MoMask, T2M-GPT) in `Table 18` in the Appendix.
4.  Included Algorithms 1 & 2 (DAgger & Diffusion details) and t-SNE visualizations in the `Appendix`.
5.  Clarified the definition of Fréchet Motion Distance (FMD) to replace the previous FID terminology.
6.  Incorporated discussions on LeVERB [6] and UH-1 [7] in the `Related Work` section.
7.  Added the ablation studies of diffusion policy in the Appendix, including noise scale, sampling strategy, and supervision target. (`Table 15, 16, 17`).
8.  Modified some typos.

All revisions are marked in yellow in the updated manuscript. We are encouraged by the consensus on our work's efficiency and novelty, and we address specific concerns below.

[1] General Motion Tracking for Humanoid Whole-Body Control

[2] Advanced Expressive Humanoid Whole-Body Control

[3] Executing your Commands via Motion Diffusion in Latent Space

[4] MoMask: Generative Masked Modeling of 3D Human Motions

[5] T2M-GPT: Generating Human Motion from Textual Descriptions with Discrete Representations

[6] LeVERB: Humanoid Whole-Body Control with Latent Vision-Language Instruction

[7] Learning from Massive Human Videos for Universal Humanoid Pose Control

---

### Author Response · Authors · 2025-12-01
**Rebuttal Summary for Paper 184**

Dear ACs and SACs,

We appreciate the reviewers’ recognition of the core contributions of our work, particularly the novelty of our retargeting-free framework, the effectiveness of the latent-driven diffusion policy, and the successful real-world deployment on the Unitree G1 humanoid.
We are encouraged by the positive recognition of our work across several key dimensions:

1.  **Novel & Elegant Framework:** Reviewers valued our “generate-control” paradigm that eliminates error-prone retargeting steps, tightly coupling semantic intent with low-level control via motion latents (`yDs5`, `eaYv`).

2.  **Efficiency & Real-World Relevance:** The significant reduction in deployment latency (from 17.85s to 5.84s) and the demonstration of sim-to-real transfer were highlighted as compelling for practical robotics (`yDs5`, `eaYv`, `3P50`, `vQfN`).

3.  **Technically Sound Methodology:** The two-stage teacher-student architecture and the hybrid transformer-diffusion design were praised for being technically solid and well-justified (`yDs5`, `eaYv`, `3P5o`).

4.  **Broad Applicability:** The framework’s potential to extend to multimodal inputs (text, image, audio) was recognized as a strong foundation for future systems (`vQfN`).

We are grateful for these positive assessments. At the same time, reviewers raised insightful questions.

To directly address these concerns, we conducted substantial additional experiments and expanded analyses, including:

1.  **State-of-the-Art Baseline Comparisons:** Implemented and evaluated GMT [1] and ExBody2 [2] on our dataset to provide a direct, head-to-head performance comparison.

2.  **Detailed Inference Latency Analysis:** Provided a module-wise breakdown of inference time and conducted a fine-grained ablation on DDIM sampling steps (2 to 10 steps) to justify the trade-off between speed and tracking accuracy.

3.  **Motion Generator Ablations:** Compared our learned latent space against pretrained motion generators, including MLD [3], MoMask [4], and T2M-GPT [5], under both frozen and fine-tuned settings to validate the necessity of our custom generator.

4.  **MoE & Architecture Validation:** Conducted ablations comparing our MoE-based teacher against 6-layer transformer to justify the architectural choice for generalization.

5.  **Algorithm & Visualization Details:** Added pseudocode for the DAgger training and Diffusion process, along with t-SNE visualizations of the motion latent space.

**Summary of Revisions:**

1.  Integrated ExBody2 and GMT comparisons in `Table 1` (Main Paper) and `Table 13` (Appendix).
2.  Included DDIM sampling step ablations (latency vs. success rate) in `Table 14` in the Appendix.
3.  Added the comparison with pretrained motion generators (MLD, MoMask, T2M-GPT) in `Table 18` in the Appendix.
4.  Included Algorithms 1 & 2 (DAgger & Diffusion details) and t-SNE visualizations in the `Appendix`.
5.  Clarified the definition of Fréchet Motion Distance (FMD) to replace the previous FID terminology.
6.  Incorporated discussions on LeVERB [6] and UH-1 [7] in the `Related Work` section.
7.  Added the ablation studies of diffusion policy in the Appendix, including noise scale, sampling strategy, and supervision target. (`Table 15, 16, 17`).
8. Added two pseudo-codes of policy training and diffusion training in the Appendix.
9.  Modified some typos.


All revisions are marked in yellow in the updated manuscript. Since most reviewers did not engage in further discussions, we believe we have now addressed every question raised by each reviewer and completed all the quantitative experiments, qualitative experiments, ablation studies, and discussions on related work requested by each reviewer. We appreciate the efforts of all Area Chairs and reviewers.

Thank you again for your hard work!

Sincerely,

The Authors of Paper 184

### References

[1] General Motion Tracking for Humanoid Whole-Body Control

[2] Advanced Expressive Humanoid Whole-Body Control

[3] Executing your Commands via Motion Diffusion in Latent Space

[4] MoMask: Generative Masked Modeling of 3D Human Motions

[5] T2M-GPT: Generating Human Motion from Textual Descriptions with Discrete Representations

[6] LeVERB: Humanoid Whole-Body Control with Latent Vision-Language Instruction

[7] Learning from Massive Human Videos for Universal Humanoid Pose Control

---

### Meta-Review · Area_Chair_5uZ2 · 2025-12-29

**Summary:**

The paper ”From Language to Locomotion: Retargeting-free Humanoid Control via Motion Latent Guidance” proposes a retargeting-free framework for language-conditioned humanoid motion control, which aims to address inference latency and error accumulation in conventional multi-stage pipelines. The approach consists of (1) generating a language-conditioned motion latent that captures semantics, (2) training a MoE-based teacher policy using privileged information and reference motion, and (3) distilling a diffusion-based student policy that directly produces robot actions conditioned on the motion latent. Experiments demonstrate reduced inference time and competitive or improved tracking performance compared to explicit retargeting baselines, with successful deployment on a real Unitree G1 humanoid robot.

**Reviewer Concerns:**

A central methodological question raised by multiple reviewers is whether the proposed framework truly grounds language semantics in robot control. Many reviewers questioned the strength of text–motion alignment at the policy level by noting that language supervision is mediated through generated motion latents rather than enforced by an explicit text–action alignment objective. Closely related to this, reviewers emphasized that the teacher policy is still trained on retargeted motion data; the student policy might not be able to meaningfully overcome retargeting-induced biases. Additional concerns focused on the justification of key design choices, including the necessity of a MoE-based teacher and diffusion-based student compared to simpler alternatives. Finally, reviewers pointed out limitations in evaluation, including reliance on retargeted ground-truth for tracking metrics and the lack of broader end-to-end comparisons and qualitative analyses to fully substantiate the claimed advantages.

**Reviewer Scores:**

From an AC perspective, the authors provided thorough responses to the reviewers’ questions, and in several cases substantially strengthened the paper through additional experiments and clarifications. In particular, concerns about missing baselines, diffusion sampling latency, MoE justification, and evaluation metrics were addressed with new results, which likely would have positively influenced reviewers had they been able to engage more fully in the discussion. One reviewer (eaYv) explicitly noted an intention to lower their score from 8 to 6, but the tone of the discussion remained constructive rather than strongly negative, and their concerns were focused on design justification rather than fundamental flaws. Other reviewers may increase their scores to compensate for that anyway. While I am personally not fully convinced that the proposed approach is completely motion-retargeting-free in a strict conceptual sense, I do agree that the work represents a solid and well-executed contribution that many researchers in the community would find useful and worth building upon.

---

### Decision · Program_Chairs · 2026-01-26

Accept (Poster)